# Satellite-based Near-Real-Time Global Daily Terrestrial Evapotranspiration Estimates

Lei Huang[1*], Yong Luo[1*], Jing M. Chen[2,3], Qiuhong Tang[4,5], Tammo Steenhuis[6], Wei Cheng[7] and Wen Shi[1]

[1]Department of Earth System Science, Ministry of Education Key Laboratory for Earth System Modeling, Institute for Global Change Studies, Tsinghua University, Beijing 100084, China
[2]Key Laboratory for Humid Subtropical Ecogeographical Processes of the Ministry of Education, School of Geographical Sciences, Fujian Normal University, Fuzhou, 350007, China
[3]Department of Geography and Planning, University of Toronto, Ontario, M5S 3G3, ON, Canada
[4]Key Laboratory of Water Cycle and Related Land Surface Processes, Institute of Geographic Sciences and Natural Resources Research, Chinese Academy of Sciences, Beijing 100101, China
[5]University of Chinese Academy of Sciences, Beijing 101408, China
[6]Department of Biological and Environmental Engineering, Cornell University, Ithaca 14850, New York, USA
[7]Key Laboratory of Land Surface Pattern and Simulation, Institute of Geographic Sciences and Natural Resources Research, Chinese Academy of Sciences, Beijing 100101, China

*Correspondence to* Lei Huang (leihuang007@mail.tsinghua.edu.cn) or
Yong Luo (Yongluo@mail.tsinghua.edu.cn)

**Abstract.**

Accurate and timely global evapotranspiration (ET) data is crucial for agriculture, water resource management, and drought forecasting. Although numerous satellite-based ET products are available, few offer near-real-time data. For instance, products like NASA's ECOSTRESS and MOD16 face challenges such as uneven coverage and delays exceeding one week in data availability. In this study, we refined the Variation of the Standard Evapotranspiration Algorithm (VISEA) by fully integrating satellite-based data, including European Centre for Medium-Range Weather Forecasts ERA5-Land's shortwave radiation, which includes satellite remote sensing data within its assimilation system and MODIS's land surface data include surface reflectance, temperature/emissivity, land cover, vegetation indices, and albedo as inputs. It enables VISEA to provide near-real-time global daily ET estimates with a maximum delay of one week at a resolution of 0.05°. Its accuracy was assessed globally using observation data from 149 flux towers across 12 land cover types and comparing it with five other satellite-based ET products and GPCC precipitation data. The results indicate that VISEA provides comparable accuracy ET estimates to existing products, achieving a mean correlation coefficient (R) of about 0.6 and an RMSE of 1.4 mm day$^{-1}$. Furthermore, we demonstrated VISEA's utility in drought monitoring during a drought event in the Yangtze River Basin in 2022, in which the ET changes correlated with the precipitation. The near-real-time capability of VISEA is, thus, especially valuable in meteorological and hydrological applications for coordinating drought relief efforts. The VISEA ET dataset is available at https://doi.org/10.11888/Terre.tpdc.300782 (Huang, 2023a).

## 1 Introduction

Global terrestrial evapotranspiration (ET) is a vital component of the Earth water cycle and energy budget. It includes evaporation from the soil and water surfaces (some studies also consider evaporation from the intercepted precipitation in canopies) and plant transpiration (He et al., 2022; Wang et al., 2021a;

Zhang et al., 2021). Accurate and timely estimation of ET is essential for quantitatively assessing changes in the water cycle under climate change, vigilant monitoring drought, and effectively managing and allocating water resources (Aschonitis et al., 2022; Han et al., 2021; Su et al., 2020).

Near-real-time ET estimation from reanalysis data has been widely used to assess ET changes in the global water cycle under different climate changes (Copernicus Climate Change Service, 2020). While these datasets, such as ERA5 (Albergel et al., 2012; Jarlan et al., 2008; Miller et al., 1992) and CRA-40 (Liu et al., 2023; Zhao et al., 2019), offer near-real-time latent heat flux (ET in energy units) with a delay of just six days, but they typically feature coarser spatial resolutions, often 0.25° or more. This level of resolution may limit their effectiveness for detailed assessments of drought conditions and the optimization of water resource allocation. On the other hand, obtaining highly accurate, near-real-time, or real-time ET measurements through local eddy covariance or lysimeter methods can be very valuable (Awada et al., 2022), but collecting large-scale ET data on a fine grid using this equipment is prohibitively expensive  (Barrios et al., 2015; Tang et al., 2009).

Satellite remote sensing-based ET estimates outperform reanalysis data by providing high spatial resolution for detailed water utilization analysis, near-real-time data for prompt environmental response, and global coverage for comprehensive water cycle studies. These ET estimates rely on direct observations, enhancing accuracy, especially where ground data are sparse, and allowing for the dynamic monitoring of land and vegetation changes.

The selected ET products discussed below have significantly contributed to estimating global ET and have gained recognition within the scientific community. The MOD16 ET product, developed by Mu et al. (2007, 2011), utilizes a Penman-Monteith-based approach and is driven by MODIS land cover, albedo, fractional photosynthetically active radiation, leaf area index, and daily meteorological reanalysis data from NASA's Global Modelling and Assimilation Office (GMAO) to estimate ET. The AVHRR ET product, developed by Zhang et al. (2006, 2009), significantly advanced the study of the global water cycle. It employed a modified Penman–Monteith approach over land, integrating biome-specific canopy conductance determined by NDVI, and utilized a Priestley–Taylor approach over water surfaces. These algorithms were driven by AVHRR Global Inventory Modeling and Mapping Studies (GIMMS) NDVI, daily surface meteorology data from the National Centers for Environmental Prediction/National Center for Atmospheric Research (NCEP/NCAR) reanalysis, and solar radiation from NASA/GEWEX Surface Radiation Budget Release-3.0. The FLUXCOM framework has made a substantial contribution to resolving the evapotranspiration paradox. It utilizes machine learning to integrate eddy covariance data from the global FLUXNET tower network, surface meteorological data from the Climatic Research Unit (CRU) reanalysis, and remote sensing data (Jung et al., 2009, 2010, 2019). Additionally, GLEAM, developed by Miralles et al. (2011b) and Martens et al. (2017), is one of the best satellite-based ET products using unique algorithmic approaches that have advanced the estimation of global ET which uses meteorology data from ECMWF Reanalysis 5. Lastly, PML, developed by Zhang et al. (2019, 2022) is the first to offer global ET coverage at a 500-meter resolution, demonstrating high accuracy compared to local eddy covariance observations worldwide with MODIS satellite data and Global Land Data Assimilation System Version 2.1 (GLDAS-2.1) data (Zhang et al., 2023).

However, these ET products cannot provide near-real-time data due to reliance on local ground-based meteorology and land-surface/reanalysis models, which are time-consuming to obtain globally. For example, MOD16 and PML use GMAO and GLDAS-2.1 data, respectively. While AVHRR ET depends on AVHRR satellite data and NCEP/NCAR Reanalysis meteorology data, GLEAM ET uses MODIS satellite data and ECMWF meteorology Reanalysis data. FLUXCOM relies on FLUXNET and the Climatic Research Unit (CRU) reanalysis data, which are not updated in real-time. Recently, NASA's ECOsystem Spaceborne Thermal Radiometer Experiment, mounted on the International Space Station on the Space Station (ECOSTRESS), was designed to estimate global-scale ET (Fisher et al., 2019, 2020). thermal infrared data at 70-meter resolution every 1 to 7 days. This results in uneven global coverage and reduced data frequency, especially in regions like the Middle East, as noted by Anderson et al., 2021 and Jaafar et al., 2022. In contrast, the VISEA model uses only MODIS land products and ERA5-Land shortwave radiation, enabling near-real-time ET estimations.

The objective of this manuscript is twofold: 1) adapt the VISEA model for near real-time, global application by replacing land-based solar radiation inputs with hourly shortwave radiation data from ECMWF ERA5-Land's data assimilation system (Sabater, 2019); and 2) to globally validate the model using a comprehensive set of datasets, including meteorological instrument data and eddy covariance measurements from 149 FLUXNET flux towers (Pastorello et al., 2020). Additionally, multi-year ET datasets from GLEAM (Martens et al., 2017; Miralles et al., 2011), FLUXCOM (Jung et al., 2009, 2010, 2018), AVHRR (Zhang et al., 2009, 2010), MOD16 (Mu et al., 2007, 2011), PML (Zhang et al., 2019, 2022) and precipitation data from the Global Precipitation Climatology Centre (GPCC) (Udo et al., 2011) are also employed in the assessment.

## 2. Methods

### 2.1 Description of the VISEA algorithm

VISEA, short for the Variation of the Moderate Resolution Imaging Spectroradiometer Standard Evapotranspiration Algorithm, is a modification of the MODIS standard Evapotranspiration (ET) algorithm. The original MODIS algorithm, created by Mu et al. (2007 and 2011), is based on the Penman-Monteith method. VISEA introduces two significant modifications. First, it employs the Vegetation (VI)-Temperature (Ts) Triangle method, originally developed by Nishida et al. (2003), to estimate air temperature. Second, VISEA incorporates hourly data on shortwave downward radiation from the ERA5-Land dataset to calculate daily average energy. These two advancements enable VISEA to estimate large-scale ET without needing local measurements as supplementary data.

Unlike energy budget-based ET algorithms, such as SEBS (Surface Energy Balance System), METRIC (Mapping Evapotranspiration at high Resolution with Internalized Calibration), and ALEXI (Atmosphere-Land Exchange Inverse), which calculate ET (latent heat flux) as the residual of the net radiation by subtracting soil heat flux and sensible heat flux. VISEA estimates ET using the Penman-Monteith equation, placing it in a different category of satellite-based global ET products currently in use. VISEA is a two-source model, which means the ET in one grid cell was separated as the transpiration from full vegetation cover and the evaporation from bare soil surface if energy transfer from the vegetation to the soil surface was ignored (Nishida et al., 2003), i.e.,

$$ET = f_{veg}ET_{veg} + (1 - f_{veg})ET_{soil} \tag{1}$$

where the subscript "$veg$" means full vegetation cover and the subscript "$soil$" indicates the soil exposed to solar radiation (called bare soil); $ET_{veg}$ is the transpiration from full vegetation cover area (W m$^{-2}$), $ET_{soil}$ is the evaporation from bare soil (W m$^{-2}$), $f_{veg}$ is the portion of the area with the vegetation cover, which can be calculated by Normalized Difference Vegetation Index (calculation details are provided in Appendix A, Tang et al., 2009).

The available energy $Q$ (W m$^{-2}$), which is the sum of the latent heat flux and sensible heat flux (also known as the net radiation minus soil heat flux) is also separated into the available energy for vegetation transpiration, $Q_{veg}$ (W m$^{-2}$) and $Q_{soil}$ (W m$^{-2}$) for bare soil evaporation, which was expressed by Nishida et al. (2003) as:

$$Q = f_{veg}Q_{veg} + (1 - f_{veg})Q_{soil} \tag{2}$$

As satellites like Terra and Aqua only provide instantaneous snapshot observations of the Earth, a temporal scaling method is needed to convert instantaneous measurements into daily ET values. Nishida et al. (2003) used satellite-based noon time instantaneous evaporation fraction ($EF$), defined as the ratio of latent heat flux ($ET$) to available energy as daily $EF$ ($EF = \frac{ET}{Q}$), multiplied the daily $Q$ to calculated daily $ET$ based on the assumption that $EF$ is constant over a day:

$$ET = EF\,Q \tag{3}$$

In the next section, we will detail how VISEA calculates the daily $EF$, and $Q$ in Eq. 3, daily air temperature and daily land surface temperature.

**2.1.1 Daily evaporation fraction calculation**

Combining Eq. 1, 2 and 3, we calculated the instantaneous evaporation fraction, $EF^i$ as:

$$EF^i = f_{veg}\frac{Q_{veg}^i}{Q^i}\,EF_{veg}^i + (1 - f_{veg})\frac{Q_{soil}^i}{Q^i}\,EF_{soil}^i \tag{4}$$

$EF_{veg}^i$ and $EF_{soil}^i$ are the instantaneous full vegetation coverage and bare soil $EF$, respectively. $EF_{veg}^i$ can be expressed as a function of instantaneous parameters (Nishida et al., 2003):

$$EF_{veg}^i = \frac{\alpha\,\Delta^i}{\Delta^i + \gamma(1 + r_{c\,veg}^i/2r_{a\,veg}^i)} \tag{5}$$

where α is the Priestley-Taylor parameter, which was set to 1.26 for wet surfaces (De Bruin, 1983); $\Delta^i$ is the instantaneous slope of the saturated vapor pressure, which is a function of the temperature (Pa K$^{-1}$); $\gamma$ is the psychometric constant (Pa K$^{-1}$); $r_{c\,veg}^i$ is the instantaneous surface resistance of the vegetation canopy (s m$^{-1}$); $r_{a\,veg}^i$ is the instantaneous aerodynamics resistance of the vegetation canopy (s m$^{-1}$). $EF_{soil}^i$ was expressed by Nishida et al. (2003) as a function of the instantaneous soil temperature and the available energy based on the energy budget of the bare soil:

$$EF_{soil}^i = \frac{T_{soil\,max}^i - T_{soil}^i}{T_{soil\,max}^i - T_a^i} \frac{Q_{soil0}^i}{Q_{soil}^i} \qquad (6)$$

where $T_{soil\,max}^i$ is the instantaneous maximum possible temperature at the surface reached when the land surface is dry (K), $T_{soil}^i$ is the instantaneous temperature of the bare soil (K), $T_a^i$ is the instantaneous air temperature, $Q_{soil0}^i$ is the instantaneous available energy for bare soil when $T_{soil}^i$ is equal to $T_a^i$ (W m$^{-2}$).

As the assumption of noon time instantaneous evaporation fraction $EF^i$ equals daily average evaporation fraction, $EF^d$, thus, $EF^i = EF^d$, caused a 10%-30% underestimation of daily ET (Huang et al., 2017; Yang et al., 2013), we introduced a decoupling parameter to covert $EF^i$ into $EF^d$ (Huang et al., 2021; Tang et al., 2017; Tang and Li, 2017). The superscript "$d$" means daily and "$i$" means instantaneous. This new decoupling parameter-based evaporation faction is developed from Penman-Monteith and McNaughton-Jarvis mathematical equations:

$$EF^d = EF^i \frac{\Delta^d}{\Delta^d + \gamma} \frac{\Delta^i + \gamma}{\Delta^i} \frac{\Omega^{*i}}{\Omega^{*d}} \frac{\Omega^d}{\Omega^i} \qquad (7)$$

where $\Omega$ is the decoupling factor that represents the relative contribution of radiative and aerodynamic terms to the overall evapotranspiration (Tang and Li, 2017), $\Omega_i^*$ is the value of the decoupling factor, $\Omega$, for wet surfaces. According to Pereira (2004), the calculation details of $\Omega$ and $\Omega^*$ are presented in Appendix B.

For full vegetation-covered areas, the decoupling parameter based daily $EF_{veg}^d$ is expressed as:

$$EF_{veg}^d = \frac{\alpha\,\Delta^i}{\Delta^i + \gamma\left(1 + \frac{r_{c\,veg}^i}{2 r_{a\,veg}^i}\right)} \left(\frac{\Delta^d}{\Delta^d + \gamma} \frac{\Delta^i + \gamma}{\Delta^i} \frac{\Omega_{veg}^{*i}}{\Omega_{veg}^{*d}} \frac{\Omega_{veg}^d}{\Omega_{veg}^i}\right) \qquad (8)$$

where $r_{c\,veg}^i$ is the instantaneous canopy resistance (s m$^{-1}$), $r_{a\,veg}^i$ is the instantaneous aerodynamic resistance (s m$^{-1}$). Determining these resistances are presented in Appendix C. For bare soil, the decoupling parameter based daily $EF_{soil}^d$ is calculated as:

$$EF_{soil}^d = \frac{T_{soil\,max}^i - T_{soil}^i}{T_{soil\,max}^i - T_a^i} \frac{Q_{soil\,0}^i}{Q_{soil}^i} \left(\frac{\Delta^d}{\Delta^d + \gamma} \frac{\Delta^i + \gamma}{\Delta^i} \frac{\Omega_{soil}^{*i}}{\Omega_{soil}^{*d}} \frac{\Omega_{soil}^d}{\Omega_{soil}^i}\right) \qquad (9)$$

Thus, $EF^d$ is expressed as:

$$EF^d = f_{veg} \frac{Q_{veg}^i}{Q^i} EF_{veg}^d + (1 - f_{veg}) \frac{Q_{soil}^i}{Q^i} EF_{soil}^d \qquad (10)$$

The same energy balance equations are used for calculating both instantaneous values $Q^i$, $Q_{veg}^i$ and $Q_{soil}^i$ and daily values $Q^d$, $Q_{veg}^d$ and $Q_{soil}^d$ but with parameters adjusted for each timeframe. The details of the calculation for the daily values are outlined below.

**2.1.2 Daily calculation of available energy $Q_{veg}^d$ and $Q_{soil}^d$**

We used an improved daily available energy $Q$ (W m$^{-2}$) method (Huang et al., 2023b) for the vegetation and the bare soil surface is calculated by the energy balance equation:

$$R_n - G = Q \tag{11}$$

where $R_n$ is the net radiation (W m$^{-2}$), which could be calculated by the land surface energy balance; $G$ is the soil heat flux (W m$^{-2}$), $G \approx 0$ on a daily basis (Fritschen and Gay, 1979; Nishida et al., 2003; Tang et al., 2009),

$$R_n^d = (1 - albedo^d)R_d^d - \varepsilon_s^d \sigma T_s^{d\,4} + (1 + Cloud^d)\varepsilon_a^d \sigma T_a^{d\,4} \tag{12}$$

where $albedo^d$ is the daily albedo of the soil surface; $R_d^d$ is daily incoming shortwave radiation (W m$^{-2}$), obtained from the ERA5_Land shortwave radiation (called ERA5_Rd); $\varepsilon_s^d$ and $\varepsilon_a^d$ are the daily emissivity of land surface and atmosphere, $\sigma$ is the Stefan-Boltzmann constant; $T_a^d$ is the daily near-surface air temperature (K); $T_s^d$ is the daily surface temperature (K). The difference with the former study by Huang et al. (2021) is that $\varepsilon_s^d$ and $\varepsilon_a^d$ were not set equal. Instead we calculated the $\varepsilon_a^d$ using the method of Brutsaert, (1975) and Wang and Dickinson(2013), as detailed in Appendix D and $\varepsilon_s^d$ can be was retrieved from MOD11C1.

We account for the influence of clouds by assuming a linear correlation between downward longwave radiation and cloud coverage in the calculation of downward longwave radiation based on the study of Huang et al. (2023b):

$$Cloud^d = (1 - K_t) \tag{13}$$

where $Cloud^d$ is the daily clearness index and $K_t$ is  (Chang and Zhang, 2019; Goforth et al., 2002)

$$K_t = \frac{R_d^d}{R_a^d} \tag{14}$$

where  $R_a^d$ is the daily extraterrestrial radiation calculated by the FAO (1998).

$Q_{veg}^d$ can be calculated by assuming as $T_s^d = T_a^d$ according to the VI-Ts method which implies that the minimum land surface temperature occurs in fully vegetated grid cells and is equivalent to $T_a^d$ (Huang et al., 2023b). According to the land surface energy budget, the daily available energy of vegetation coverage area, $Q_{veg}^d$ and bare soil $Q_{soil}^d$ can be calculated following the study of Huang et al. (2023b):

$$Q_{veg}^d = (1 - albedo^d)R_d^d + (1 + Cloud^d)\varepsilon_a^d \sigma T_a^{d\,4} - \varepsilon_s^d \sigma T_s^{d\,4} \tag{15}$$

$$Q_{soil}^d = (1 - C_G)(1 - albedo^d)R_d^d + (1 + Cloud^d)\varepsilon_a^d \sigma T_a^{d\,4} - \varepsilon_s^d \sigma T_s^{d\,4} \tag{16}$$

The daily mean air temperature, $T_a^d$ can be extended by a sin and cos function based on the instantaneous air temperature $T_a^i$ which was calculated using the linear correlation between vegetation

index (VI) and surface temperature (Ts) method. Thus, $(1 + Cloud^d)\varepsilon_a^d \sigma T_a^{d\,4}$ is the daily downward
longwave radiation (W m$^{-2}$), and $\varepsilon_s^d \sigma T_s^{d\,4}$ is the daily upward longwave radiation (W m$^{-2}$), where $C_G$ is
an empirical coefficient ranging from 0.3 for a wet soil to 0.5 for a dry soil (Idso et al., 1975).
$Q_{veg}^d$ and $Q_{soil}^d$ are calculated by the energy balance equations, which are robust on both
instantaneous and daily scales. Thus instantaneous $Q_{veg}^i$ and $Q_{soil}^i$ are calculated by the same set of
equsing Eq 17 andd 18 by replacing the daily by the instantaneous parameters.
Following the study of Huang et al. (2023b), the daily $ET^d$ can be calculated by the daily $EF^d$ and
$Q^d$ as:

$$ET^d = EF^d Q^d \tag{17}$$

Figure 1 illustrates the workflow of VISEA. VISEA utilizes land cover data from the MOD12C1
IGBP land cover classification. When land cover in a MOD12C1 IGBP data grid cell is identified as a
water surface, VISEA then uses the Priestley-Taylor equation to compute water surface evaporation. This
process guarantees that the unique attributes of water surfaces are precisely reflected in VISEA ET
calculations.

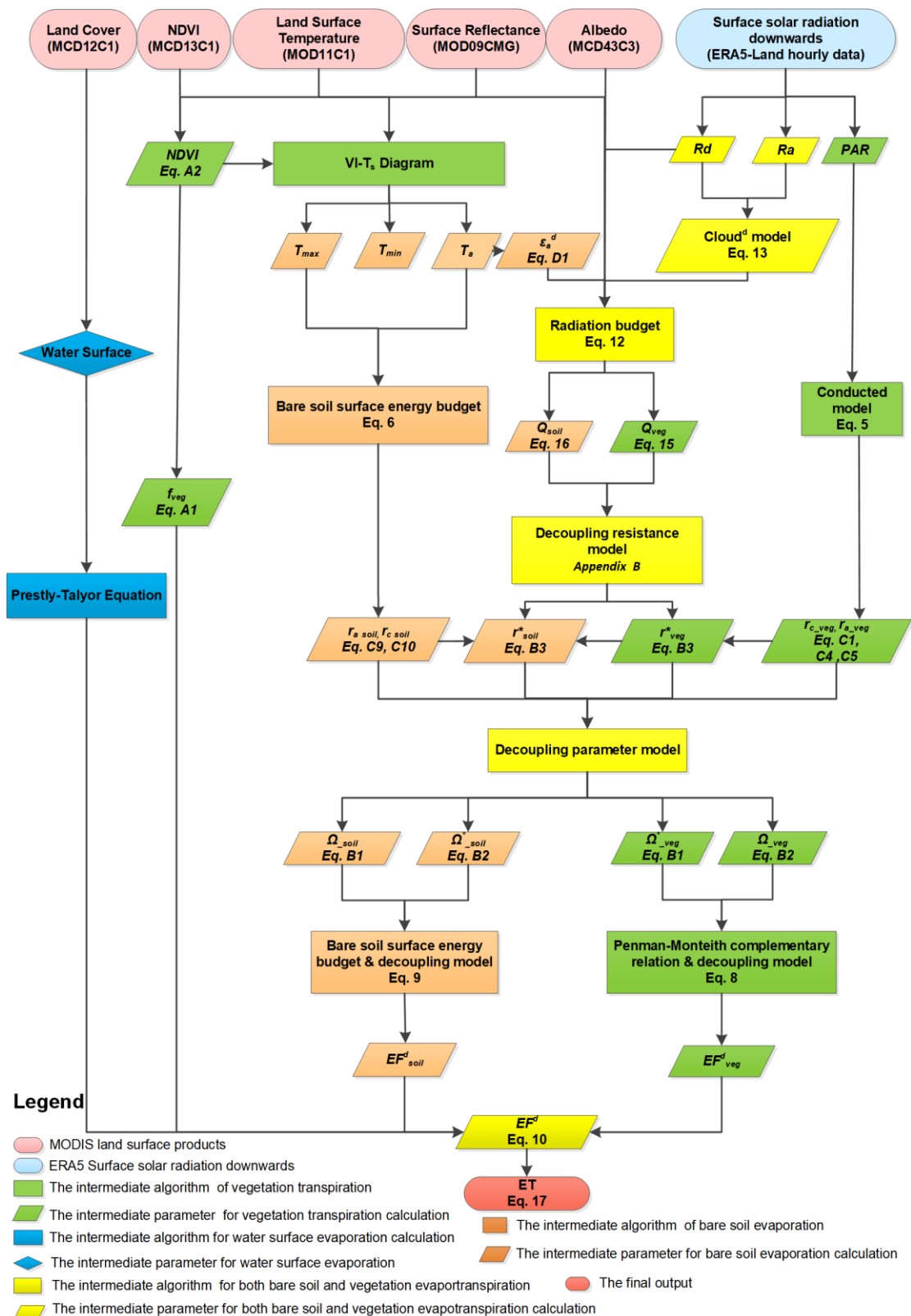

**Figure 1.** Schematic of VISEA algorithm. The ovals in the top row are the databases, the square boxes are the algorithms, and parallelograms are the parameters. The numbers in the parenthesis are the equation to determine the parameters.

### 2.1.3 The calculation of daily air temperature, $T_a^d$ and surface temperature, $T_s^d$

Daily air temperature, $T_a^d$ is a critical parameter in the VISEA algorithm, used in calculations for downward longwave radiation, daily aerodynamic resistance, and surface resistance. The key innovation in calculating $T_a^d$, involves employing the VI-Ts method to estimate instantaneous air temperature, $T_a^i$ during the daytime (Huang et al., 2017; Nishida et al., 2003; Tang et al., 2009).

This VI-Ts method was developed based on the empirical linear relationship between the surface temperature (Ts) and the Vegetation Index (VI). Surface temperature increases when the vegetation index decreases, and conversely, surface temperature decreases when the vegetation index increases. In the scatter plot, defined by VI (horizontal axis) and Ts (vertical axis) from the neighboring $5 \times 5$ grid cells, we identify the "warm edge" (characterized by a low vegetation cover fraction and high Ts) and the "cold edge" (marked by a high vegetation cover fraction and low Ts). The warm edge is automatically selected as the hypotenuse of the triangle formed by these scatter points. Through simple interpolation, Ts corresponding to any given vegetation condition within the range of the "warm edge" and "cold edge" can be determined. The lowest Ts could be determined by the highest VI, and the highest Ts could be determined by the lowest VI. Therefore, following Nishida et al. (2003), assuming that the lowest surface temperature equals the air temperature (Ta), we can derive the daily air temperature.

For nighttime periods, it is assumed that air temperature is equivalent to the nighttime land surface temperature provided by MOD11C1. These two temperature estimates are then extended into hourly air temperature profiles using a sine-cosine fitting curve. The 24-hour average of $T_a^i$ is used as $T_a^d$. Similarly, $T_s^d$ is calculated using MOD11C1 land surface temperature data for both daytime and nighttime. These estimates are extended into hourly surface temperature profiles using a similar sine-cosine fitting curve, and the daily average of $T_s^d$ is determined (Huang et al., 2021).

This VI-Ts method allows for the estimation of $T_a^i$ and $T_{soil\ max}^i$ without the need for additional meteorological data. However, some studies have found that the VI-Ts method may not consistently provide satisfactory results, especially in colder regions where vegetation thrives better under higher temperatures.

### 2.2 Technical validation

The correlation coefficient, Root Mean Square Error (RMSE) and Nash-Sutcliffe efficiency coefficient are used to evaluate our global daily ET estimates with eddy covariance measurements and compared with the other five independent global ET products on a monthly scale.

The correlation coefficient R is calculated as:

$$R = \frac{\sum (X - \bar{X})(Y - \bar{Y})}{\sqrt{\sum (X - \bar{X})^2 \sum (Y - \bar{Y})^2}} \tag{18}$$

$R$ is the correlation coefficient; $X$ is the estimated variable; $\bar{X}$ is the average of $X$; $Y$ is the observed variable; $\bar{Y}$ is the average of $Y$.

The Root Mean Square Error (RMSE) is calculated as:

$$RMSE = \sqrt{\frac{\sum_{i=1}^{N}(X_i - Y_i)^2}{N}}$$

262                                                                                                          (19)

For a more nuanced understanding of the Root Mean Square Error (RMSE), we have deconstructed
it into two distinct components: RMSEs (systematic RMSE) and RMSEu (unsystematic RMSE). This
breakdown allows a more detailed examination of the systematic and unsystematic sources contributing
to the overall error metric.
The systematic Root Mean Square Error (RMSEs) is calculated as:

$$RMSEs = \sqrt{\frac{\sum_{i=1}^{N}(Z_i - Y_i)^2}{N}}$$

268                                                                                                          (20)

The unsystematic Root Mean Square Error (RMSEu) is calculated as:

$$RMSEu = \sqrt{\frac{\sum_{i=1}^{N}(Z_i - X_i)^2}{N}}$$

270                                                                                                          (21)

Where $Z_i = a + bY_i$, where a and b are the least squares regression coefficients of the estimated variable
$X_i$ and observed variable $Y_i$, $N$ is the sample size (Norman et al., 1995).
The Nash-Sutcliffe efficiency coefficient (NSE)

$$NSE = 1 - \frac{\sum(X_i - Y_i)^2}{\sum(Y_i - \bar{Y})^2}$$

274                                                                                                          (22)

The ratio of the standard deviations of $X$ and $Y$

$$Ratio = \frac{X_{Standard\ Deviation}}{Y_{Standard\ Deviation}}$$

276                                                                                                          (23)

The Bias of $X$ and $Y$

$$Bias = \bar{X} - \bar{Y}$$

278                                                                                                          (24)

**2.3 The gap-filling of MODIS data**
MODIS sensors on board of Terra and Aqua observe the Earth twice a day. However, there are
always data gaps in the MODIS land products because of cloud cover problems. In the VISEA algorithm,
we used the data from the neighboring days to fill the data gaps. The periods when MODIS Land
temperature data were missing, primarily due to cloud cover, accounted for approximately one-third of
the observation period. The accuracy of this gap-filling method is evaluated in Section 4.

## 3. Data

### 3.1 The input data

The input data including the MODIS land products: daily 0.05° surface reflectance (MOD09CMG), land surface temperature/emissivity (MOD11C1) and albedo (MCD43C3), 8-day 0.05° vegetation indices (MOD13C1) and yearly 0.05° land cover products (MCD12C1). We also used hourly downward surface solar radiation from the Fifth Generation of the European Centre for Medium-Range Weather Forecasts (ECMWF) Reanalysis (ERA5), "ERA5-Land hourly data from 1950 to present" data as energy input of VISEA algorithm. The surface solar radiation data from ERA5-Land and land data products from MODIS land products are both near-real-time datasets with a one-week delay, enabling VISEA to provide global near-real-time ET estimations. Details of the input data, their download links, variable names, used parameters, spatial and temporal resolution are given in Table 1.

**Table 1. The input of VISEA**

| The input of VISEA | | | |
|---|---|---|---|
| Data source | Data name | Used parameter | Spatial/temporal resolution |
| **MODIS Land Product** | MOD11C1 | Land Surface Temperature | 0.05°/ daily |
| | MOD09CMG | Surface Reflectance | 0.05°/daily |
| | MCD43C3 | Albedo | 0.05°/daily |
| | MOD13C1 | NDVI | 0.05°/16-day |
| | MCD12C1 | Land Cover | 0.05°/ yearly |
| **ERA5-Land hourly data** | Rd | Downward Surface Solar Radiation | 0.1°/ hourly |

### 3.2 The evaluation data

### 3.2.1 The flux tower measurements from FLUXNET

We evaluated the accuracy of the input ERA5-Land shortwave radiation, estimated daily net radiation, air temperature, and ET by comparing them against measurements from FLUXNET2015 (Pastorello et al., 2020). FLUXNET consists of 212 globally distributed flux towers and it has implemented quality control measures for energy closure and is considered reliable (Baldocchi et al., 2001; Pastorello et al., 2020; Wang et al., 2022). The data from FLUXNET2015 can be obtained at https://fluxnet.org/data/download-data. We selected data from 2001 to 2015 and excluded sites with zero ERA5-Land downward shortwave radiation.

While there are records from 212 flux towers in our datasets, not all met the stringent inclusion criteria. Each site needed to fulfill three specific requirements to be included in our analysis: (1) availability of data for the period spanning from 2001 to 2015; (2) ERA5-Land downward shortwave radiation greater than 0 within the $0.1° \times 0.1°$ grid cell corresponding to the flux tower's location; (3) conformity with MODIS land cover data (MOD12C1) at the $0.05° \times 0.05°$ grid cell level, ensuring that the flux tower was situated on land rather than over the ocean. Based on these criteria, we selected a

subset of 149 flux towers that met these stringent criteria. This approach ensures the reliability and relevance of our analysis. The distribution of these 149 flux towers is presented in Figure 2. Supplementary Table S1 shows the longitude, latitude, elevation, and land cover type (classified by the International Geosphere-Biosphere Programme, IGBP) of these sites. The 149 sites covered 12 IGBP land cover types: 18 croplands (CRO), 1 closed shrublands (CSH), 15 deciduous broadleaf forests (DBF), 1 deciduous needle leaf forest (DNF), 10 evergreen broadleaf forests (EBF), 34 evergreen needle leaf forests (ENF), 30 grasslands (GRA), 5 mixed forests (MF), 8 open shrublands (OSH), 8 savannas (SAV), 13 wetlands (WET), and 6 woody savannas (WSA).

**3.2.2 The other gridded ET and precipitation products**

Five independent globally gridded ET products and one precipitation product were used to evaluate VISEA estimated ET. The five ET products include two MODIS-based ET products: MOD16 (Mu et al., 2007, 2011) and Penman-Monteith-Leuning Evapotranspiration V2 (PML) (Zhang et al., 2019, 2022), one AVHRR-based AVHRR ET (Zhang et al., 2009, 2010), one machine learning algorithm output, the FLUXCOM ET data (Jung et al., 2009, 2010, 2018, 2019) and one multiple-satellites data based Global Land Evaporation Amsterdam Model (GLEAM) ET (Martens et al., 2017; Miralles et al., 2011). The precipitation data was from the Global Precipitation Climatology Centre (GPCC), which is based on local measurements (Becker et al., 2013; Schneider et al., 2014, 2017) and Global Unified Gauge-Based Analysis of Daily Precipitation (GPC). Details of these five ET products and the precipitation data are given in Table 2. To maintain the consistency in temporal and spatial resolution for comparison purposes, we obtained monthly MOD16 and PML despite their original temporal resolution of 8 days. We used the $0.05°\times0.05°$ version of MOD16, AVHRR ET and PML. Additionally, for multi-year scale comparisons, we confined our dataset to the timeframe between 2001 and 2020. This selection enabled us to utilize a diverse range of ET products, effectively minimizing the influence of temporal discrepancies on our comparative analysis. We also incorporated daily Evapotranspiration (ET) data from GLEAM and VISEA, alongside precipitation data from the Climate Prediction Center (CPC), from July 25[th] to August 2[nd], 2022. It allowed for near-real-time analysis of ET and precipitation during the Yangtze River drought incident within that interval, despite the datasets potentially encompassing more extensive periods.

**Table 2.** The five global girded ET products and one precipitation product used for comparison with our near-real-time global daily terrestrial ET estimates.

| Product name | Spatial/Temporal resolution | Time period | Theory |
|---|---|---|---|
| GLEAM | 0.25°/Monthly | 2001-2022 | Priestly-Taylor Equation |
| FLUXCOM | 0.5°/Monthly | 2001-2016 | Machine Learning |
| MOD16 | 0.05°/Monthly | 2001-2014 | Penman-Monteith Equation |
| AVHRR | 1°/Monthly | 2001-2006 | Improved Penman-Monteith Equation |
| PML | 0.05°/8-day | 2003-2018 | Penman-Monteith Equation and A Diagnostic Biophysical Model |
| GPCC | 0.25°/Monthly | 2001-2019 | In-situ Observations |
| GPC | 0.5°/Daily | 08/28/2022-09/01/2022 | Global Unified Gauge-Based Analysis of Daily Precipitation |

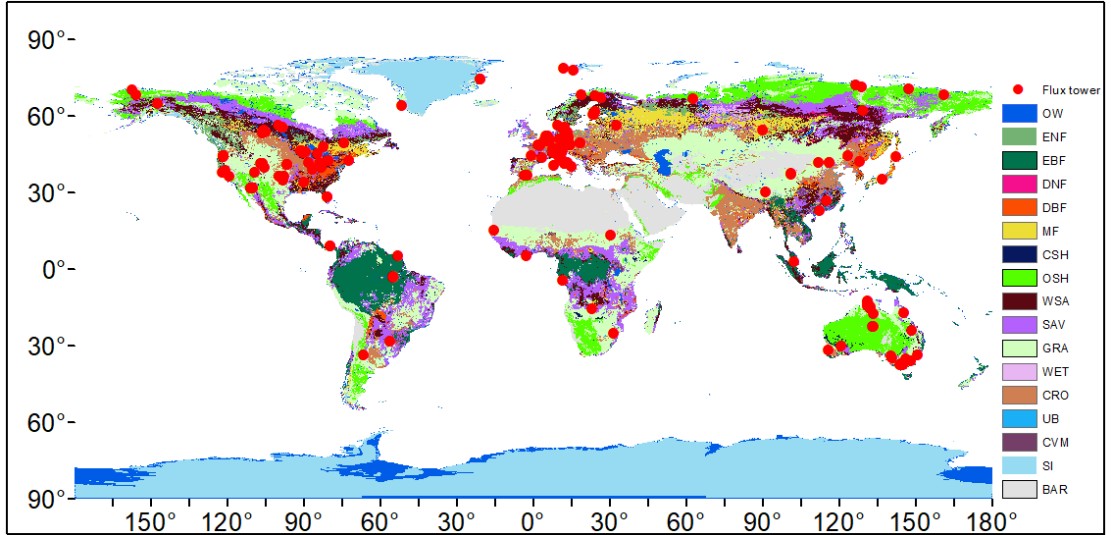

343

**Figure 2.** The distribution of 149 flux towers from FLUXNET in different IGBP land cover types, specifically OW (Water bodies), ENF (Evergreen needle leaf forests), EBF (Evergreen broadleaf forests), DNF (Deciduous needle leaf forests), DBF (Deciduous broadleaf forests), MF (Mixed forests), CSH (Closed shrublands), OSH (Open shrublands), WSA (Woody savannas), SAV (Savannas), GRA (Grasslands), WET (Permanent wetlands), CRO (Croplands), UB (Urban and built-up lands), CVM (Cropland/natural vegetation mosaics), SI (Snow and ice), BAR (Barren).

## 4. Results

To evaluate the performance of ERA5_Rd across different land cover initial categories, we juxtaposed downward solar radiation input data from ERA5-Land (ERA5_Rd) with measurements obtained from 149 flux towers (Obv_Rd) across diverse IGBP land cover types, as illustrated in Figure 3. The results indicate a commendable agreement between ERA5_Rd and Obv_Rd measurements for the majority of land covers, with notable exceptions observed in savanna (SAV). Specifically, the mean Nash-Sutcliffe Efficiency (NSE) stands at 0.84, the mean correlation coefficient (R) at 0.92, and the mean Root Mean Square Error (RMSE) at 38.3 W m$^{-2}$.

Figure 3 shows that ERA5 input shortwave radiation generally agrees well with local measurements. ERA5_Rd exhibits optimal performance in DNF and MF, reflected by NSE and R values surpassing 0.9. In these land covers, the mean RMSEs stand at 11 W m$^{-2}$, mean RMSEu at 24.5 W m$^{-2}$, and mean RMSE at 26.9 W m$^{-2}$. However, its performance in SAV is notably subpar, characterized by an NSE of 0.29, an R of 0.59, highest RMSEs of 40 W m$^{-2}$, RMSEu of 48.9 W m$^{-2}$, and RMSE of 63.2 W m$^{-2}$. For ERA5_Rd, the mean RMSEs amount to 16 W m$^{-2}$, and the mean RMSEu is 34.8 W m$^{-2}$, suggesting that ERA5_Rd demonstrates high accuracy by effectively capturing the systematic variation in Obv_Rd, as indicated by its relatively low RMSEs and RMSEu close to RMSE (Willmott et al., 1981) in most land covers, except for SAV. Specifically, in Figure 3, Rd s derived from ERA5 exhibit very low P-values (<0.01).

367

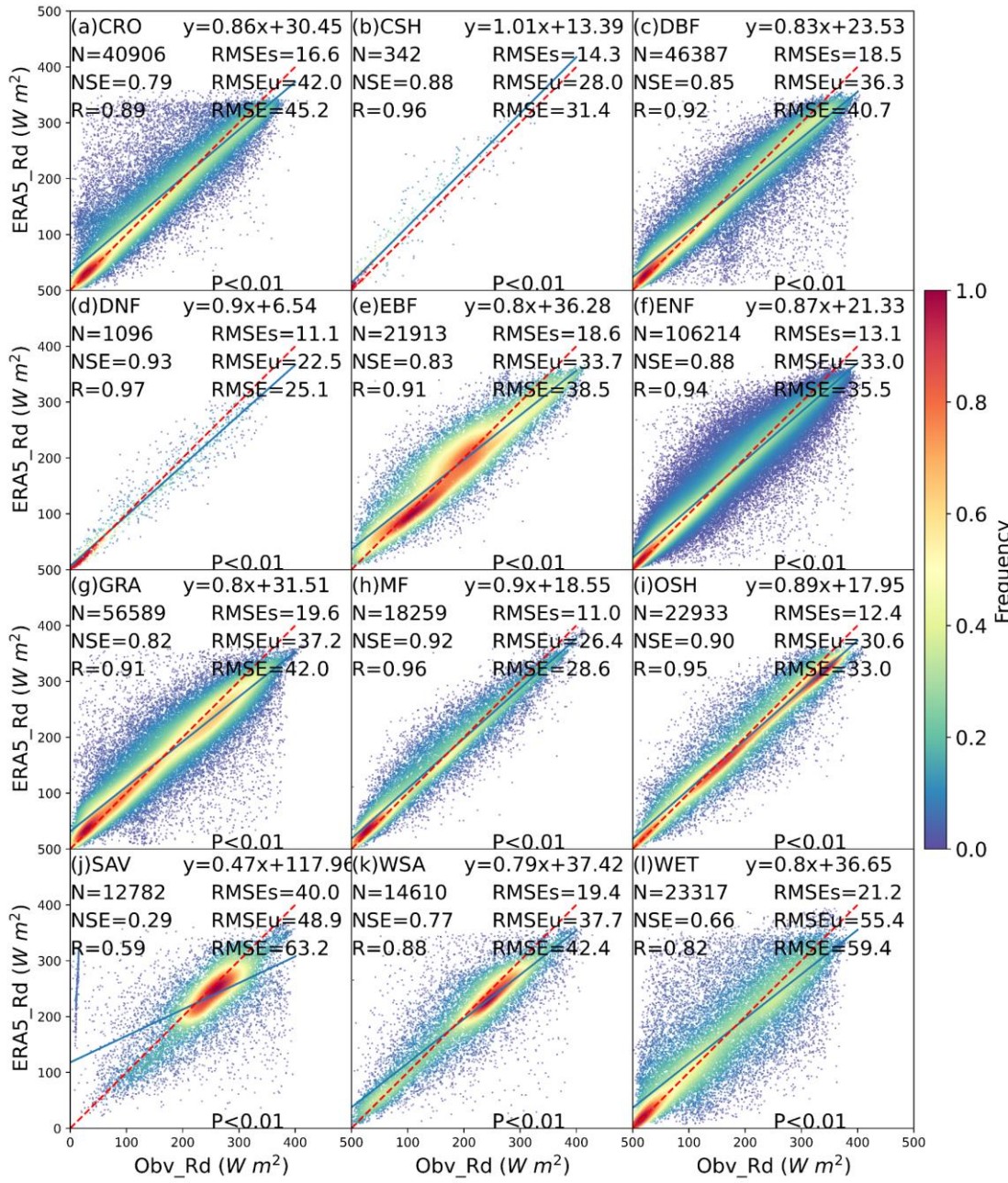

368

**Figure 3.** The scatter plot of downward solar radiation from ERA5-Land (ERA5_Rd) compared with
local instruments measurements (Obv_Rd) under 12 IGBP land cover types: CRO (Croplands), CSH
(Closed shrublands), DBF (Deciduous broadleaf forests), DNF (Deciduous needle leaf forests), EBF
(Evergreen broadleaf forests), ENF (Evergreen needle leaf forests), GRA (Grasslands), MF (Mixed
forests), OSH (Open shrublands), SAV (Savannas), WSA (Woody savannas), WET (Permanent
wetlands). The red dotted line is the 1:1 line. N is the number of data points, NSE is Nash-Sutcliffe
Efficiency, R is correlation coefficients, RMSE is Root Mean Square Error, RMSEs is systematic RMSE,
and RMSEu is unsystematic RMSE. The Frequency denotes the probability density estimated through
the KDE method with a Gaussian kernel, and it is then scaled to ensure that the maximum value of the
probability density function equals 1. P is the P-Value for the Correlation Coefficient.

Several factors come into play in understanding the disparities in performance in downward solar
radiation of ERA5 (ERA5_Rd) across different land cover types. In regions characterized by denser
forests, such as DNF and MF, ERA5_Rd's good performance may be attributed to the lower density of
ground-based meteorology stations (DNF, N = 1096) and the relatively uniform subsurface and canopy
coverage in MF, facilitating a more accurate representation in the ERA5 radiative transfer model.
Conversely, savannas present unique challenges due to sparse vegetation and flat terrain, influencing
sunlight transmission dynamics (Yang and Friedl, 2003). Land-use changes, including farming and urban
development, further complicate the accuracy of sunlight transmission. Additionally, factors like aerosols
from natural or anthropogenic sources contribute to data variations (Naud et al., 2014; Wang et al.,
2021b). The inaccuracies in accounting for the rainy season, leading to increased cloud cover and rainfall
in savannas, contribute to ERA5_Rd's limitations (Jiang et al., 2020).
We chose to utilize 0.05° MODIS data for its detailed land surface information, daily time step, and
global coverage, which is essential for accurate and near-real-time ET calculations. Although ERA5 data
is at a coarser 0.1° resolution, it provides necessary atmospheric inputs that can be effectively interpolated
to match the MODIS resolution without significant loss of accuracy. As illustrated in Figures 3 and 4,
our tests confirm that this method achieves accurate ET despite the resolution differences.
Figure 4 depicts scatter plots illustrating the comparison between the estimated air temperature using
the VI-$T_S$ method (VISEA_Ta) and local meteorological measurements (Obv_Ta). The analysis reveals
that VISEA_Ta generally aligns with Obv_Ta, exhibiting NSE values ranging from -0.22 (MF) to 0.82
(OSH), R values ranging from 0.44 (MF) to 0.97 (DNF), and RMSE values ranging from 5.7 K (WSA)
to 11.2 K (MF). Particularly noteworthy is VISEA_Ta's outstanding performance at OSH (NSE = 0.82,
R = 0.93, RMSE = 6.6 K), WSA (NSE = 0.79, R = 0.92, RMSE = 5.7 K) and GRA (NSE = 0.66, R =
0.88, RMSE = 6.8 K). Conversely, the least satisfactory performance is evident at MF (NSE = -0.22, R
= 0.44, RMSE = 11.2 K), SAV (NSE = -0.19, R = 0.57, RMSE = 6.4 K), and CRO (NSE = 0.26, R =
0.70, RMSE = 8.1 K). The RMSEs are lower than RMSEu in most land cover sites, except in DNF.
Despite VISEA_Ta displaying a high NSE of 0.8 and R of 0.97 at DNF, it exhibits higher RMSEs (8.3
K) compared to RMSEu (5.4 K), indicating a systematic underestimation of VISEA_Ta at DNF.
As detailed in Section 2.4, the VI-Ts method relies on a negative correlation between vegetation
coverage (VI) and land surface temperature (Ts), ideally suited for cases with significant VI and Ts
differences. However, the assumed negative correlation breaks down for land cover types like DNF and
MF in temperate regions with distinct seasons and cool-to-cold climates. In these regions, the positive
correlation between VI and Ts, driven by vegetation growth proportional to rising Ts, results in the failure
of the VI-Ts method. The challenges persist in SAV, where the VI-Ts method encounters difficulties
during dry and wet seasons. In the dry season, the method falters due to the prevalence of bare soil,
resulting in VI values approaching zero and homogeneous high Ts values. Conversely, the wet season
presents challenges, with both VI and Ts exhibiting relatively high values and limited variances between
grid cells, ultimately undermining the accuracy of VISEA_Ta estimation.

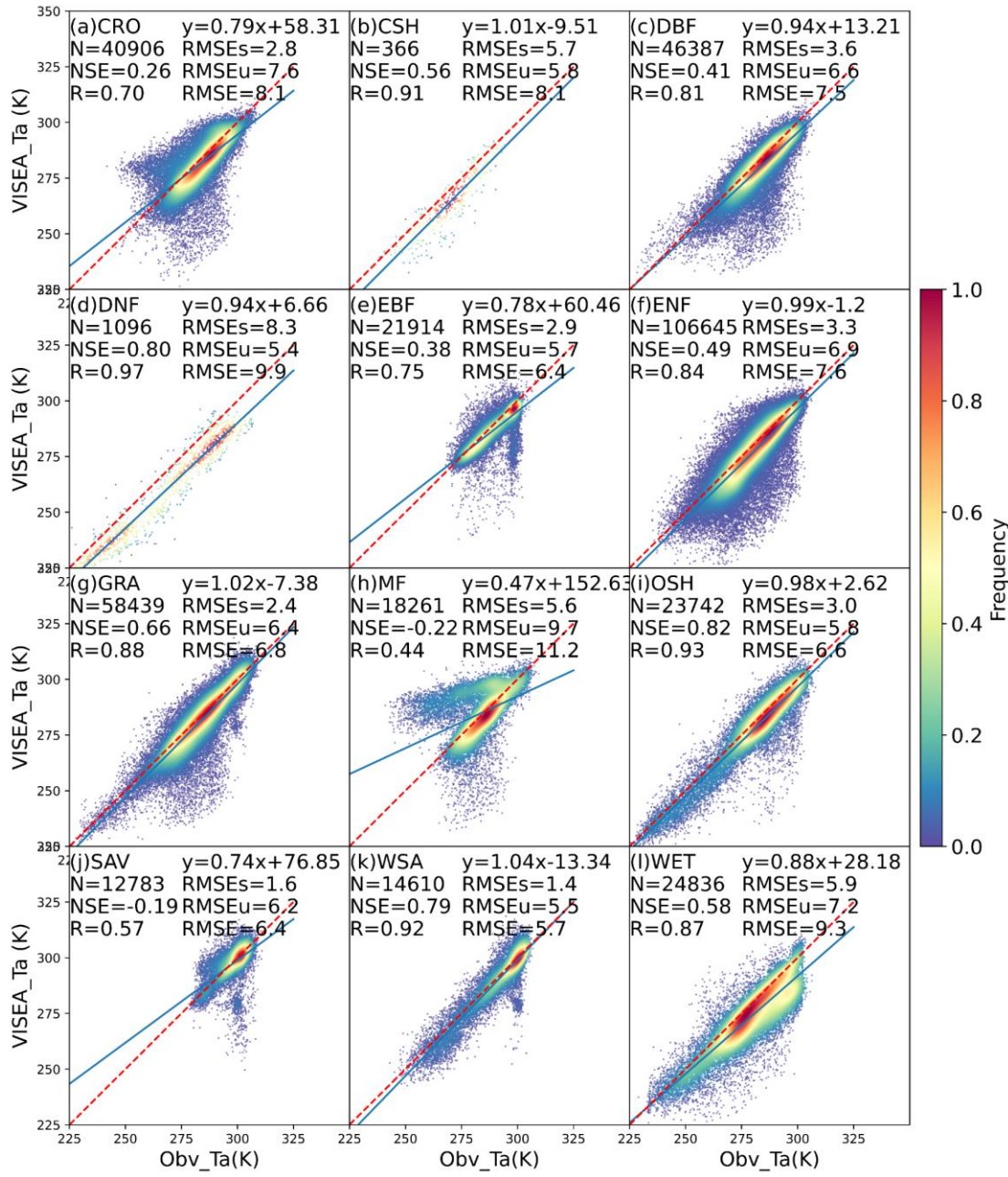

**Figure 4.** The scatter plot of daily air temperature simulated by VISEA (VISEA_Ta) compared with local instruments measurements (Obv_Ta) under 12 IGBP land cover types: CRO (Croplands), CSH (Closed shrublands), DBF (Deciduous broadleaf forests), DNF (Deciduous needle leaf forests), EBF (Evergreen broadleaf forests), ENF (Evergreen needle leaf forests), GRA (Grasslands), MF (Mixed forests), OSH (Open shrublands), SAV (Savannas), WSA (Woody savannas), WET (Permanent wetlands). The red dotted line is the 1:1 line. N is the number of data points, NSE is Nash-Sutcliffe Efficiency, R is correlation coefficients, RMSE is Root Mean Square Error, RMSEs is systematic RMSE, and RMSEu is unsystematic RMSE. The frequency denotes the probability density estimated through the Kernel Density Estimation, KDE method with a Gaussian kernel, and it is then scaled to ensure that the maximum value of the probability density function equals 1.

The simulated daily net radiation (VISEA_Rn) from VISEA is assessed against local meteorological
measurements (Obv_Rn) in Figure 5. In contrast to the satisfactory performance of ERA5_Rd in Figure
3, VISEA_Rn exhibits more notable discrepancies, characterized by significant underestimation
compared to Obv_Rn. This is reflected in the mean NSE of 0.49, mean R of 0.74, and mean RMSE of
43.3 W m$^{-2}$. Specifically, VISEA_Rn demonstrates good accuracy in certain land cover types, including
CHS with an NSE of 0.67, R of 0.84, and RMSE of 29.7 W m$^{-2}$, EBF with an NSE of 0.63, R of 0.8, and
RMSE of 42.9 W m$^{-2}$, and ENF with an NSE of 0.66, R of 0.83, and RMSE of 39.6 W m$^{-2}$. However, its
performance diminishes notably at OSH, where it records an NSE of 0.16, R of 0.61, and RMSE of 56
W m$^{-2}$, as well as in SAV, with an NSE of 0.21, R of 0.52, and RMSE of 44.2 W m$^{-2}$. While VISEA_Rn
appears to have lower accuracy compared to ERA5_Rd, in the majority of land cover types, the RMSEs
are smaller than RMSEu, with mean RMSEs of 25.2 W m$^{-2}$ and mean RMSEu of 34.3 W m$^{-2}$. Moreover,
the RMSEu of 43.3 W m$^{-2}$ is almost the same as the RMSE.
In the context of VISEA_Rn, a consistent pattern of approximately 30% underestimation in net
radiation across various land cover types raises noteworthy discussions. This systematic discrepancy
could be linked to the disparity in vegetation coverage between the observed sites' footprint and the mean
vegetation coverage of the 0.05° × 0.05° grid cell. Specifically, the lower albedo within the footprint,
compared to the grid cell's average albedo (as expressed by Eq. 14, contributes to the underestimation of
Obv_Rn. This is particularly evident in OSH, where the vegetation coverage within the footprint
significantly exceeds the mean vegetation coverage of the grid cell (<0.2 compared to >0.5). Factors such
as the bias in ERA5_Rd (refer to Fig. 3j) and VISEA_Ta (refer to Fig. 4j) contribute to the
underestimation of VISEA_Rn in SAV. Moreover, a substantial 50% underestimation in DNF results
from the underestimated VISEA_Ta (refer to Fig. 4d) leads to a subsequent underestimation of downward
long-wave radiation.

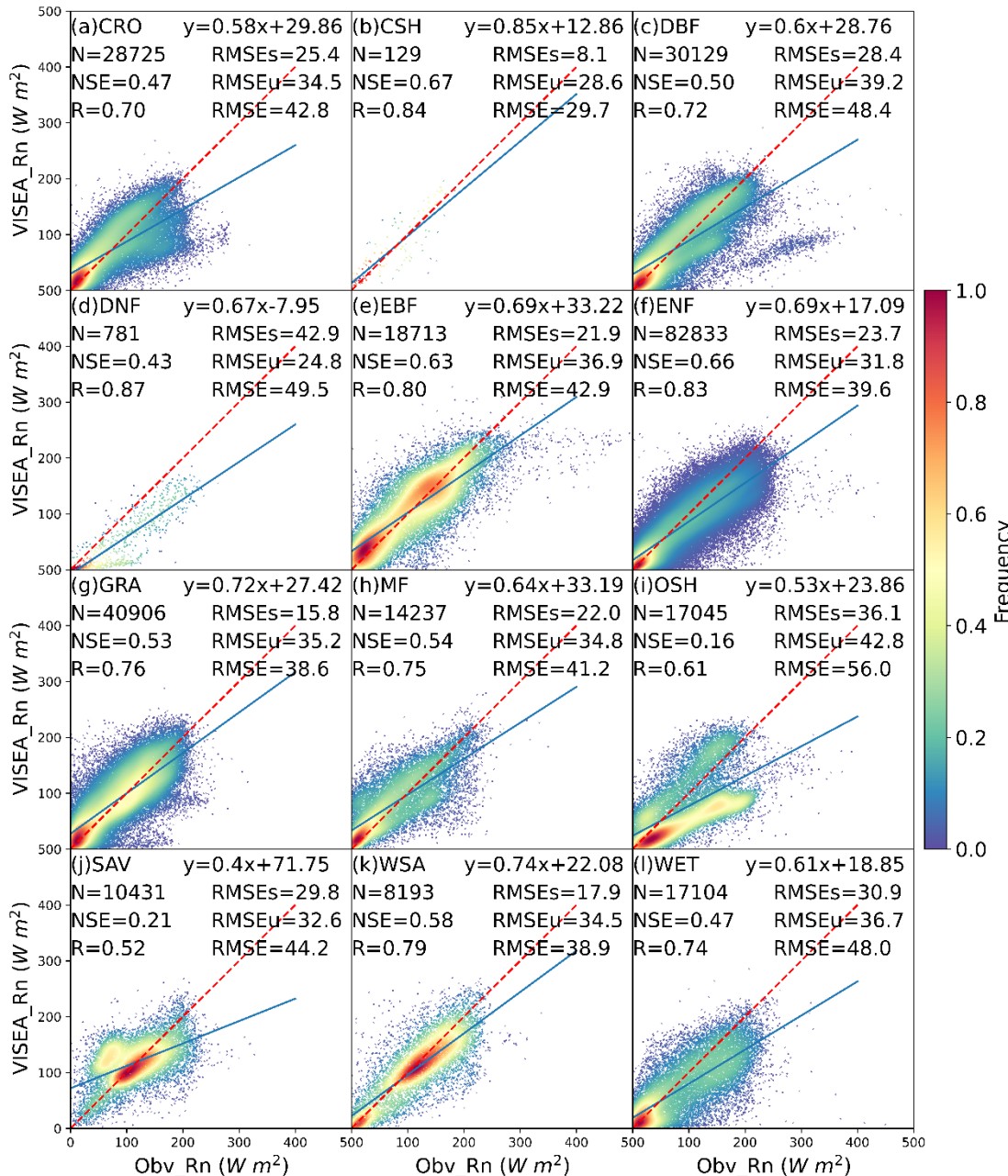


**Figure 5.** The scatter plot of daily net radiation simulated by VISEA (VISEA_Rn) compared with local instruments measurements (Obv_Rn) under 12 IGBP land cover types: CRO (Croplands), CSH (Closed shrublands), DBF (Deciduous broadleaf forests), DNF (Deciduous needle leaf forests), EBF (Evergreen broadleaf forests), ENF (Evergreen needle leaf forests), GRA (Grasslands), MF (Mixed forests), OSH (Open shrublands), SAV (Savannas), WSA (Woody savannas), WET (Permanent wetlands). The red dotted line is the 1:1 line. N is the number of data points, NSE is Nash-Sutcliffe Efficiency, R is correlation coefficients, RMSE is Root Mean Square Error, RMSEs is systematic RMSE, and RMSEu is unsystematic RMSE. The frequency denotes the probability density estimated through the Kernel Density Estimation, KDE method with a Gaussian kernel, and it is then scaled to ensure that the maximum value of the probability density function equals 1.

Figure 6 illustrates scatter plots of daily evapotranspiration (ET) simulated by VISEA (VISEA_ET)
against eddy covariance measurements obtained from 149 flux tower sites (Obv_ET) across 12 IGBP
land cover types. The scatter plots of VISEA_ET reveal a dispersed distribution, as evidenced by an
average NSE of -0.08, average R of 0.56, and average RMSE of 1.4 mm day$^{-1}$. Notably, VISEA_ET tends
to underestimate daily ET across most land cover types. Among the 12 land cover types, VISEA_ET
exhibits the highest accuracy in DNF, with an NSE of 0.4, an R of 0.82, and an RMSE of 0.9 mm day$^{-1}$.
It was closely followed by GRA, with NSE values of 0.26, R values of 0.65, and RMSE values of 1.3
mm day$^{-1}$. However, for CRO, ENF, and WET land cover types, the NSE values, although above 0, are
close to 0 (mean NSE of 0.11), with a mean R of 0.53 and a mean RMSE of 1.3 mm day$^{-1}$. In the remaining
land cover types, particularly in OSH and SAV, VISEA_ET appears to struggle in aligning with local
measurements, resulting in NSE values of -0.57 and -0.51, R values of 0.31 and 0.36, and RMSE values
of 1.2 mm day$^{-1}$ and 1.7 mm day$^{-1}$, respectively. As the evaluation of daily VISEA_ET with observed
ET, Obv_ET, at CRO and WET, the bias mainly comes from the bias in ERA5_Rd (the third highest
RMSE of 45.2 W m$^{-2}$ and second highest RMSE of 59.4 W m$^{-2}$) (Fig. 3a and l). In ENF, the biases
primarily is caused by the disability of VISEA_ET to capture the Obv_ET under a cold climate, with low
net radiation estimation (Fig. 5f) and air temperature (Fig. 4f). For OSH, the bias mainly arises from the
poor estimation of VISEA_Rn, which has the lowest NSE of 0.16 and the highest RMSE of 56 W m$^{-2}$
(Fig. 5i). The bias of VISEA_ET in SAV is a result of the combined biases in ERA5_Rd (the lowest NSE
and R of 0.29 and 0.59, respectively, and the highest RMSE of 63.2 W m$^{-2}$), VISEA_Ta (the second
lowest NSE and R of  -0.19 and 0.57, respectively).
The periods when MODIS land temperature data were missing, primarily due to cloud cover,
accounted for approximately one-third of the observation period. Using the gap-filling method (section
2.3), it can be observed that for most surfaces, the accuracy of VISEA was not significantly affected by
clouds, as evidenced by the figures below. The accuracy on cloudy days is slightly lower for some
surfaces compared to clear days. For example, in the case of DBF, the correlation coefficient R is 0.52
on both clear and cloudy days, and the RMSE is 1.4 mm day$^{-1}$ on both clear and cloudy days, indicating
a slight decrease in accuracy under cloudy conditions. Similarly, for ENF, the R value is 0.59 on clear
days and 0.56 on cloudy days. At the same time, the RMSE is 1.3 mm day$^{-1}$ on clear days and 1.4 mm
day$^{-1}$ on cloudy days, showing that although there is some impact, the overall performance of VISEA
remains robust across different weather conditions (Figures S4 and S5).

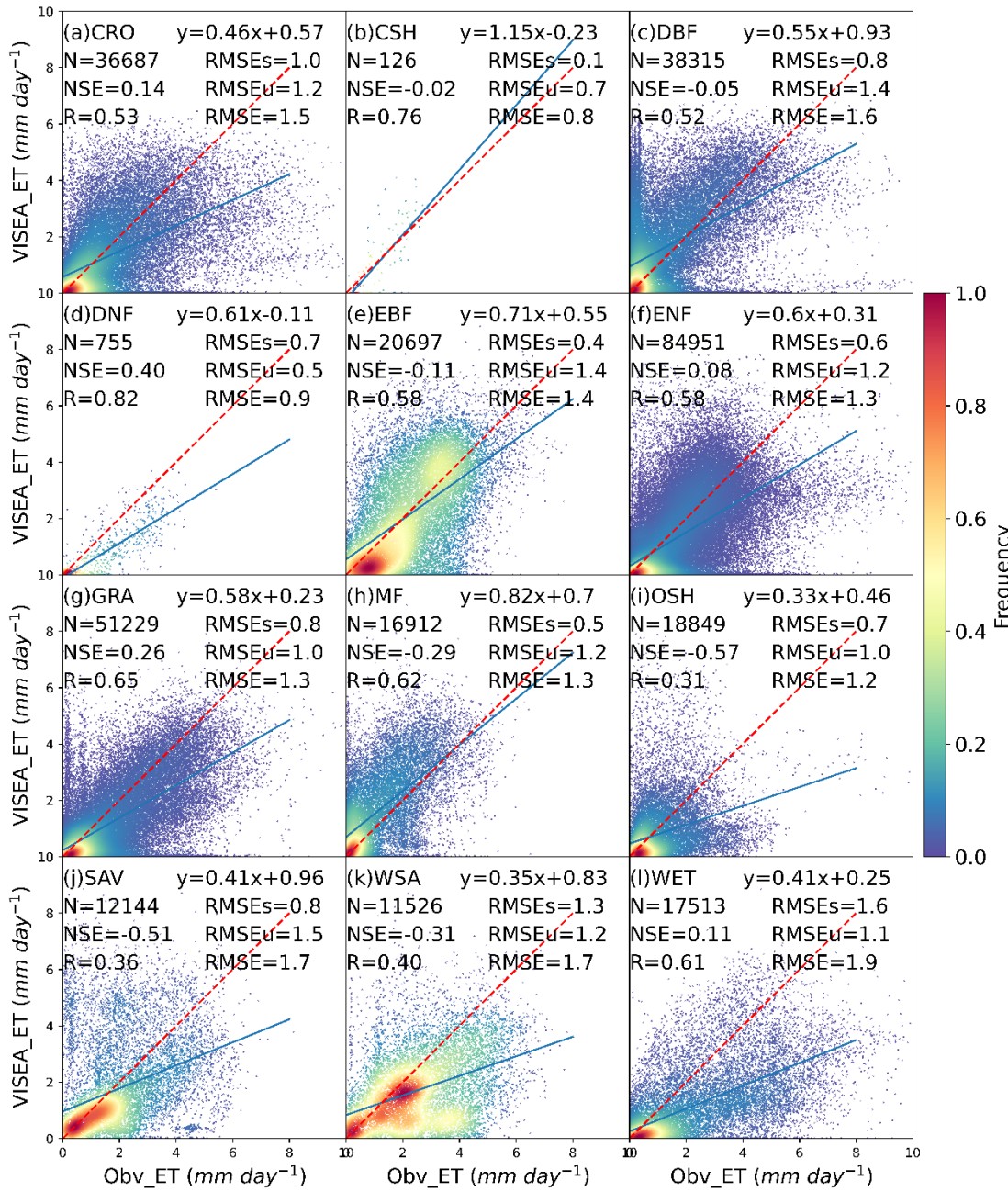


**Figure 6.** The scatter plot of daily ET simulated by VISEA (VISEA_ET) compared with local instruments measurements (Obv_ET) under 12 IGBP land cover types: CRO (Croplands), CSH (Closed shrublands), DBF (Deciduous broadleaf forests), DNF (Deciduous needle leaf forests), EBF (Evergreen broadleaf forests), ENF (Evergreen needle leaf forests), GRA (Grasslands), MF (Mixed forests), OSH (Open shrublands), SAV (Savannas), WSA (Woody savannas), WET (Permanent wetlands). The red dotted line is the 1:1 line. N is the number of data points, NSE is Nash-Sutcliffe Efficiency, R is correlation coefficients, RMSE is Root Mean Square Error, RMSEs is systematic RMSE, and RMSEu is unsystematic RMSE. The frequency denotes the probability density estimated through the Kernel Density Estimation, KDE method with a Gaussian kernel, and it is then scaled to ensure that the maximum value of the probability density function equals 1.

We also conducted the VISEA sensitivity to different radiation input data by comparing results
obtained using CERES and ERA5 datasets. Specifically, we analyzed the performance of the VISEA
model in simulating net radiation (Rn) and evapotranspiration (ET), comparing these simulations with
ground-based observational data. Figures S1 and 2 compare the downward shortwave radiation data from
CERES and ERA5 with ground-based observations of the 149 flux towers. The CERES shortwave
radiation data generally agree with the observational data, with a mean R of 0.89, a mean RMSE of 34.8
W m$^2$, and a mean NSE of 0.78. In contrast, the ERA5 shortwave radiation data mean R of 0.85, a mean
RMSE of 40.4 W m$^2$, and a mean NSE of 0.58 when compared with the ground-based observations,
indicating systematic bias and lower precision for the ERA5 net radiation compared with CERES. Figures
S2 and 5 compare the net radiation of the flux towers with that calculated by the VISEA model with
shortwave radiation of CERES and ERA5 as input data. For CERES data, the mean R is 0.74, the mean
RMSE is 34.3 W m$^2$ and the mean NSE is 0.64. The ERA5 data yield a mean R of 0.64, a mean RMSE
of 39.44 W m$^2$, and a mean NSE of 0.44. Finally, the ET calculated with the VISEA using the net radiation
of CERES and ERA5 as input is compared with ground-based data in Figures S3 and 6. Again, CERES
outperforms ERA5 as indicated by the statistical measures. The sensitivity analysis reveals that the
VISEA model's performance highly depends on the quality of the incident radiation data used as input.
The model shows better accuracy and consistency with CERES data than ERA5 data. Therefore, selecting
high-precision radiation data is crucial for improving the accuracy and reliability of VISEA model
simulations.
In Figure 7, we utilized Taylor diagrams (Taylor, 2001) to evaluate the performances of six global
gridded monthly ET products with simulated ET from VISEA (a), GLEAM (b), FLUXCOM (c), AVHRR
(d), MOD16 (e), and PML (f). Table 3 lists the statistical metrics, including correlation coefficient (CC),
bias, RMSE, RMSEu, RMSEs, and Nash-Sutcliffe Efficiency (NSE) across different vegetation types
and their mean values. The vegetation types include Croplands (CRO), Closed Shrublands (CSH),
Deciduous Broadleaf Forest (DBF), Deciduous Needleleaf Forest (DNF), Evergreen Broadleaf Forest
(EBF), Evergreen Needleleaf Forest (ENF), Grasslands (GRA), Mixed Forests (MF), Open Shrublands
(OSH), Savannas (SAV), Woody Savannas (WSA), Wetlands (WET), and an overall mean (MEAN).
VISEA, with a mean correlation coefficient (CC) of 0.69, indicates moderate correlation across
vegetation types but suffers from significant biases, notably in WET, with a mean bias of -9.56 mm
month$^{-1}$. It also has the highest mean Root Mean Square Error (RMSE) at 31.6 mm month$^{-1}$ and a mean
NSE of 0.25. MOD16 demonstrates a slightly better correlation with a mean CC of 0.72 and presents less
variation in bias, resulting in a marginally lower mean RMSE of 28.7 mm month$^{-1}$ and a higher mean
NSE of 0.36. AVHRR matches VISEA in mean CC at 0.69 but exhibits extreme biases, particularly in
SAV, and achieves a comparable mean RMSE of 26.3 mm month$^{-1}$. However, its mean NSE of 0.10 is
the lowest among the six products, suggesting its predictions are less reliable.
On the other hand, GLEAM, FLUXCOM, and PML show better agreements. GLEAM has a high
mean CC of 0.69 with the lowest bias at -0.82 mm month$^{-1}$, indicating consistent performance with a
mean RMSE of 29.6 mm month$^{-1}$ and a mean NSE of 0.31. FLUXCOM exhibits a higher mean CC of
0.76, suggesting better overall correlation, but with a higher mean bias of 6.2 mm month$^{-1}$, it hints at a
tendency towards overestimation. The mean RMSE is 30.0 mm month$^{-1}$, with a mean NSE of 0.22. PML
outperforms the others, with the highest mean CC of 0.75 and the highest mean NSE of 0.49, indicating
the strongest predictive accuracy. It also has the lowest mean RMSE at 26.0 mm month$^{-1}$.

**Figure 7.** Taylor Diagrams comparing monthly measurements of (a) VISEA, GLEAM (b), FLUXCOM (c), AVHRR (d), MOD16 (e), and PML (f) with 150 flux towers (labeled as Obv) in different IGBP land cover types. The diagrams display the Normalized Standard Deviation (represented by red circles), Correlation Coefficient (shown as green lines), and Centred Root-Mean-Square (depicted as blue circles).

**Table 3.** Statistical variables of six ET Products – CC (Correlation Coefficient), Ratio (the ratio of the standard deviations of simulated ET and flux tower measurements), Bias, RMSE, RMSEu, RMSEs, and NSE.

| | | CRO | CSH | DBF | DNF | EBF | ENF | GRA | MF | OSH | SAV | WSA | WET | MEAN |
|---|---|---|---|---|---|---|---|---|---|---|---|---|---|---|
| **VISEA** | CC | 0.57 | 0.89 | 0.67 | 0.95 | 0.74 | 0.71 | 0.72 | 0.79 | 0.39 | 0.55 | 0.6 | 0.66 | 0.69 |
| | Ratio | 0.77 | 1.27 | 0.99 | 0.76 | 1.29 | 1.02 | 0.8 | 1.27 | 1.06 | 0.7 | 0.78 | 0.63 | 0.95 |
| | Bias | -14.16 | -1.27 | 3.9 | -19.06 | 1.37 | -11.15 | -13.47 | 1.53 | -6.83 | -0.45 | -23.14 | -31.98 | -9.56 |
| | RMSE | 39.4 | 12.5 | 34 | 22.1 | 30.4 | 29.3 | 32 | 23.3 | 30.4 | 32.5 | 41.2 | 51.6 | 31.56 |
| | RMSEU | 27.4 | 12.1 | 30.7 | 7.4 | 30.4 | 25.3 | 23.1 | 23.2 | 25.4 | 22.5 | 25.8 | 25.4 | 23.23 |
| | RMSES | 28.3 | 3.1 | 14.5 | 20.8 | 2.2 | 14.7 | 22.2 | 1.5 | 16.8 | 23.5 | 32.1 | 44.9 | 18.72 |
| | NSE | 0.18 | 0.64 | 0.34 | 0.45 | 0.24 | 0.3 | 0.41 | 0.38 | -0.36 | 0.28 | 0.01 | 0.08 | 0.25 |
| | | | | | | | | | | | | | | |
| **GLEAM** | CC | 0.56 | 0.94 | 0.61 | 0.89 | 0.81 | 0.67 | 0.71 | 0.81 | 0.51 | 0.53 | 0.57 | 0.67 | 0.69 |
| | Ratio | 0.7 | 1.28 | 0.79 | 0.82 | 0.99 | 1.1 | 0.78 | 1.04 | 1.12 | 0.96 | 0.95 | 0.56 | 0.92 |
| | Bias | -6.13 | 12.52 | 5.8 | -5.04 | 5.42 | 4.37 | -1.16 | 10.51 | 5.62 | -7.1 | -16.73 | -17.91 | -0.82 |
| | RMSE | 37.2 | 15.4 | 34.2 | 14.7 | 21.8 | 30.3 | 29.6 | 21.4 | 28.6 | 37.1 | 40.9 | 44.4 | 29.63 |
| | RMSEU | 25.3 | 8 | 25.9 | 11.2 | 20.1 | 28.5 | 22.8 | 18 | 25.5 | 31.2 | 32.1 | 22.5 | 22.59 |
| | RMSES | 27.2 | 13.1 | 22.3 | 9.4 | 8.6 | 10.3 | 18.8 | 11.5 | 12.8 | 20 | 25.3 | 38.3 | 18.13 |
| | NSE | 0.27 | 0.35 | 0.33 | 0.75 | 0.61 | 0.25 | 0.5 | 0.47 | -0.17 | 0.06 | 0.02 | 0.32 | 0.31 |
| | | | | | | | | | | | | | | |
| **FLUXCOM** | CC | 0.66 | 0.98 | 0.69 | 0.95 | 0.79 | 0.77 | 0.75 | 0.83 | 0.78 | 0.59 | 0.65 | 0.69 | 0.76 |
| | Ratio | 0.94 | 1.76 | 0.96 | 1.04 | 1.12 | 1.18 | 0.97 | 1.42 | 0.97 | 1.04 | 1.08 | 0.62 | 1.09 |
| | Bias | 7.22 | 23.49 | 17.57 | -2.26 | 6.29 | 7.08 | 6.91 | 21.02 | 10.04 | 0.74 | -9.75 | -14.04 | 6.19 |
| | RMSE | 35.8 | 27.9 | 36.7 | 9.9 | 25.2 | 27.7 | 30 | 31.9 | 19.8 | 35.5 | 37.8 | 41.7 | 29.99 |
| | RMSEU | 31 | 5.8 | 28.9 | 9.7 | 24.1 | 26.6 | 26.8 | 23.5 | 15.8 | 32.3 | 34.3 | 24.2 | 23.58 |
| | RMSES | 18 | 27.3 | 22.6 | 2.3 | 7.5 | 7.8 | 13.4 | 21.6 | 11.9 | 14.8 | 15.8 | 33.9 | 16.41 |
| | NSE | 0.32 | -1.14 | 0.23 | 0.88 | 0.48 | 0.38 | 0.48 | -0.17 | 0.43 | 0.14 | 0.17 | 0.4 | 0.22 |
| | | | | | | | | | | | | | | |
| **AVHRR** | CC | 0.8 | 0 | 0.8 | 0 | 0.76 | 0.67 | 0.58 | 0.79 | 0.69 | 0.32 | 0.7 | 0.79 | 0.58 |
| | Ratio | 0.91 | 0 | 0.87 | 0 | 0.87 | 1.14 | 0.83 | 0.9 | 0.89 | 0.3 | 0.95 | 0.43 | 0.67 |
| | Bias | -1.15 | 0 | 5.96 | 0 | 5.24 | -1.72 | -7.04 | 0.16 | -2.41 | -47.83 | -0.42 | -25.32 | -6.21 |
| | RMSE | 23.6 | 0 | 26.1 | 0 | 23.3 | 31.1 | 36 | 18.8 | 22.1 | 54.7 | 33.2 | 46.6 | 26.29 |
| | RMSEU | 21.2 | 0 | 22 | 0 | 19.5 | 29.9 | 27.9 | 16.6 | 18.8 | 8 | 29.8 | 14.6 | 17.36 |
| | RMSES | 10.4 | 0 | 14.1 | 0 | 12.7 | 8.5 | 22.7 | 8.7 | 11.6 | 54.2 | 14.6 | 44.2 | 16.81 |
| | NSE | 0.63 | 0 | 0.61 | 0 | 0.54 | 0.22 | 0.24 | 0.62 | 0.43 | -2.79 | 0.42 | 0.29 | 0.10 |
| | | | | | | | | | | | | | | |
| **MOD16** | CC | 0.57 | 0.94 | 0.71 | 0.95 | 0.82 | 0.73 | 0.71 | 0.81 | 0.67 | 0.53 | 0.59 | 0.65 | 0.72 |
| | Ratio | 0.64 | 1.26 | 0.77 | 0.8 | 1.11 | 0.81 | 0.74 | 1.09 | 0.66 | 1 | 1 | 0.46 | 0.86 |
| | Bias | -7.88 | 14.03 | 5.79 | -4.07 | 7.17 | -4.34 | -5.05 | 4.09 | -6.41 | -16.01 | -23.76 | -21.07 | -4.79 |
| | RMSE | 36.9 | 16.7 | 30.7 | 11.1 | 23.4 | 24.6 | 29.6 | 19.4 | 20.4 | 40.4 | 44.3 | 47.2 | 28.73 |
| | RMSEU | 23 | 8.4 | 23 | 7.4 | 22 | 19.5 | 21.7 | 18.7 | 12.8 | 32.4 | 33.3 | 18.8 | 20.08 |
| | RMSES | 28.8 | 14.4 | 20.3 | 8.2 | 7.8 | 15 | 20.2 | 5.2 | 15.9 | 24.2 | 29.1 | 43.3 | 19.37 |
| | NSE | 0.28 | 0.24 | 0.48 | 0.87 | 0.55 | 0.51 | 0.5 | 0.57 | 0.39 | -0.12 | -0.14 | 0.23 | 0.36 |
| | | | | | | | | | | | | | | |
| **PML** | CC | 0.68 | 0.99 | 0.68 | 0.93 | 0.8 | 0.79 | 0.68 | 0.77 | 0.7 | 0.57 | 0.61 | 0.82 | 0.75 |
| | Ratio | 0.8 | 1.04 | 0.81 | 1.22 | 0.98 | 0.97 | 0.79 | 0.96 | 1.01 | 0.94 | 0.83 | 0.56 | 0.91 |
| | Bias | -6.6 | -3 | 3.39 | 0.47 | -1.42 | -5.43 | -6.66 | -0.59 | 6.48 | -0.18 | -16.04 | -22.1 | -4.31 |
| | RMSE | 33.2 | 4.1 | 31.5 | 13.3 | 21.9 | 23 | 31.7 | 19.8 | 21.1 | 34.5 | 37.5 | 40.5 | 26.01 |
| | RMSEU | 25.6 | 2.8 | 25.1 | 12.7 | 20.5 | 20.8 | 24.1 | 18.2 | 18.6 | 29.5 | 27.1 | 17.3 | 20.19 |
| | RMSES | 21.1 | 3.1 | 19 | 3.9 | 7.8 | 9.6 | 20.6 | 7.7 | 9.9 | 17.8 | 26 | 36.6 | 15.26 |
| | NSE | 0.42 | 0.95 | 0.44 | 0.79 | 0.61 | 0.57 | 0.43 | 0.55 | 0.33 | 0.19 | 0.16 | 0.43 | 0.49 |

Figure 8 illustrates the spatial distribution of the multi-year average (a-g), the zonal mean (h) and inter-annual variation (i) of (a) GPCC (2001-2019), (b) VISEA (2001-2020), (c) GLEAM (2001-2020), (d) FLUXCOM (2001-2016), (e) AVHRR (2001-2006), (f) MOD16 (2001-2014) and (g) PML (2003-2018).

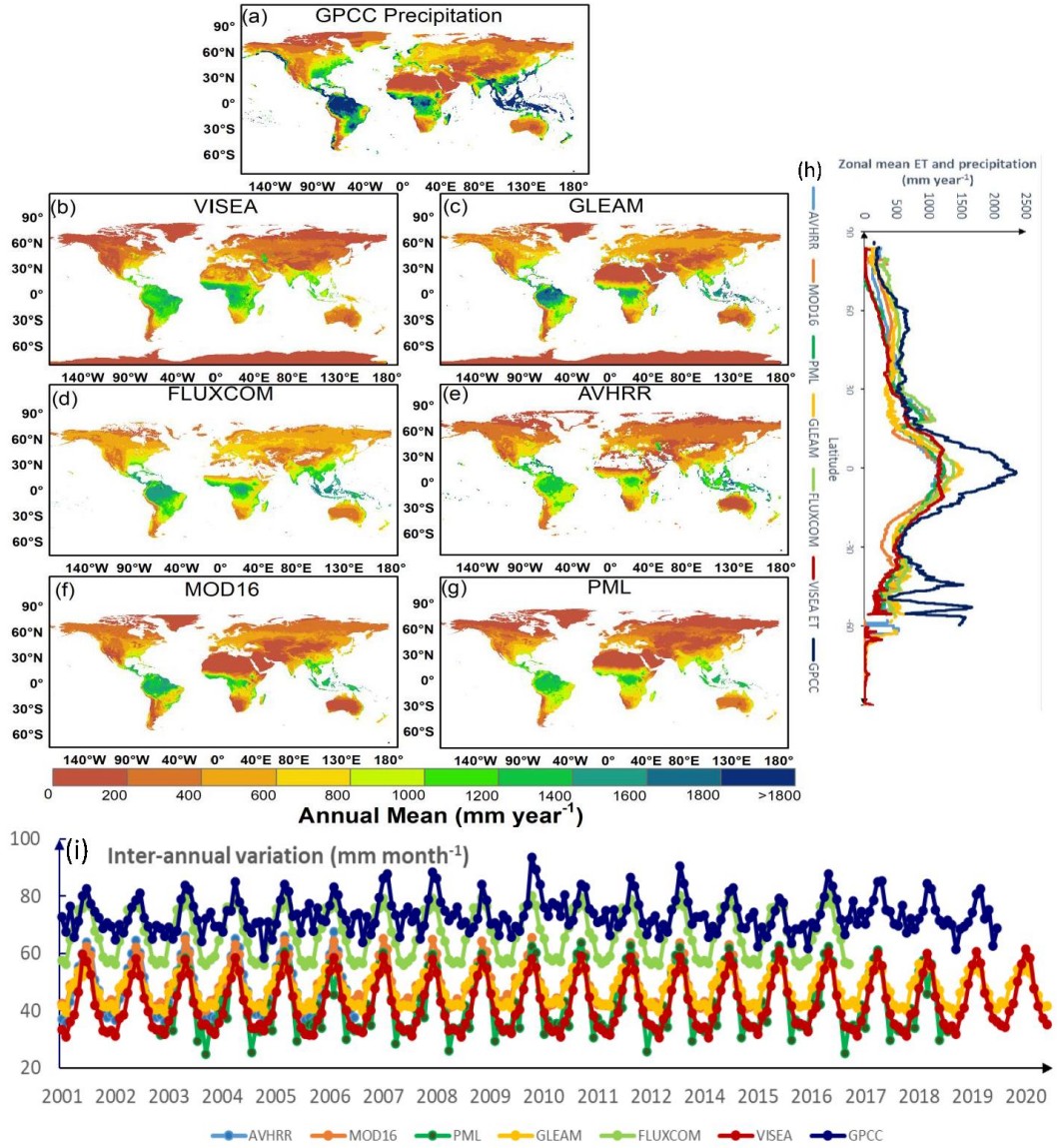

**Figure 8.** The spatial distribution of the multi-year average (a-g), the zonal mean (h) and inter-annual variation (i) of (a) GPCC precipitation (2001-2019), (b) VISEA (2001-2020), (c) GLEAM (2001-2020), (d) FLUXCOM (2001-2016), (e) AVHRR (2001-2006), (f) MOD16 (2001-2014) and (g) PML (2003-2018) ET data.

The VISEA ET product demonstrates consistent spatial distribution patterns among the six ET products across various years in terms of annual means (a-g) and latitude zonal means (h). These patterns closely align with the precipitation distribution data from GPCC. Furthermore, VISEA ET also exhibit similar spatial distributions compared to other ET products, particularly in the extremes of the distribution, below the 5th percentile and above the 95th percentile (Figure S6, S7). The highest ET values, approximately 1,500 mm year[-1], are predominantly in equatorial low-latitude regions with the corresponding high precipitation levels of approximately 2,500 mm year[-1]. These regions include South America (Amazon Basin), Central Africa (Congo Basin), and Southeast Asia (encompassing Indonesia, Malaysia, parts of Thailand, and the Philippines), which have tropical rainforest climates. Remote sensing data support the ET estimates and align with findings from previous studies, such as Chen et al. (2021)

and Zhang et al. (2019), who reported that the multi-year average annual ET is nearly 1,500 and the
precipitation is approximately 2,500 mm year$^{-1}$. Also, Panagos et al. (2017) report similar multi-year
average annual ET and precipitation rates.
In this analysis, barren lands (BAR) such as the Sahara, Arabian, Gobi, and Kalahari deserts, along
with large areas of Australia, and snow and ice (SI) regions including significant parts of Canada, Russia,
and the Qinghai-Tibet Plateau in China, are characterized by notably low evapotranspiration (ET). These
regions typically experience less than 400 mm year$^{-1}$ of annual ET, paralleled by minimal yearly
precipitation ranging from 200 to 400 mm year$^{-1}$, according to GPCC data. Comparative ET rates for
other land cover types generally range from 400 to 1,400 mm year$^{-1}$, closely following the GPCC
precipitation amounts of 600 to 1,600 mm year$^{-1}$.
In regions experiencing moisture-limited evapotranspiration (ET), the scarcity of available water is
the primary constraint. Conversely, in areas where sufficient water is available, ET is energy-limited, and
factors such as cloud cover or shading restrict the absorption of solar radiation, affecting the
evapotranspiration rate. Panel (i) in Figure 8 illustrates inter-annual monthly variations over the past two
decades. It shows how VISEA and other satellite-based ET products, alongside GPCC precipitation data,
capture the rhythmic patterns of ET. These data reveal distinctive seasonal fluctuations and highlight the
significant inter-annual climate variability. Among these products, FLUXCOM consistently shows ET
values 10-20 mm month$^{-1}$ higher than those of other ET products. GLEAM and MOD16 exhibit similar
ET estimations, closely paralleling each other, as do PML and VISEA. Notably, after 2007, both GLEAM
and MOD16 reported higher ET estimations than PML and VISEA in November, December, January,
and February. For the same months, PML consistently records lower ET estimations than VISEA.
Analysis across the datasets reveals how ET estimates respond to extreme climate events, providing
insights into the variability and resilience of these models. For instance, during the 2011-2012 drought in
the Horn of Africa—one of the most severe droughts in recent decades—both ET estimations and GPCC
precipitation data showed significant declines. Similarly, the prolonged California drought from 2012 to
2016 also saw a considerable decrease in ET values, aligning with the reduced precipitation levels
captured by GPCC.
Regarding the inter-annual monthly variations, panel (i) shows the fluctuations in ET across
different years for the analyzed ET products and precipitation data. The graph reveals a rhythmic pattern
of ET across the years. VISEA and other ET products showed distinctive peaks and troughs
corresponding to seasonal changes and inter-annual climate variability. The ET products' data align
closely with the precipitation patterns reported by GPCC, highlighting the interconnectedness between
ET and precipitation as climatic variables. Notably, FLUXCOM consistently presents higher ET
estimations than the other products. GLEAM's ET estimations are also slightly higher during the winter,
indicating a trend of systematic overestimation in these products relative to the others in the dataset.
Figure 9 presents the daily ET from VISEA and GLEAM, alongside precipitation data from the
GPCC across the Yangtze River Basin from August 26$^{th}$ to September 2$^{nd}$, 2022. During this period, a

significant drought was observed in the region, which began in July and showed signs of abating by late August and early September, according to Zhang et al. (2023). VISEA ET illustrates the evolving drought conditions, with notably low ET levels (below 1 mm day$^{-1}$) across the basin from August 26$^{th}$ to 28$^{th}$, as shown in panels (a-c). A marked increase in precipitation on August 29$^{th}$, evident in panels (s) and (u), correlates with an uptick in ET values (surpassing 1 mm day$^{-1}$) throughout the basin, visualized in panels (d-f). Although GLEAM generally captures the fluctuations in ET—both decreases and increases— during this period, it consistently reports much higher ET values than VISEA. The panel (y) graph in Figure 9 shows the precipitation and the ET calculated by VISAE and GLEAM after an 11 mm rainfall on August 29$^{th}$. The ET of VISEA increased and the deceased, which is expected because ET and soil moisture are positively correlated. The GLEAM does not follow the expected pattern shown in panel y. This comprehensive analysis highlights the interdependence of precipitation and ET and underscores the importance of considering soil moisture dynamics to fully understand the hydrological processes within the Yangtze River Basin during extreme weather events.

Beyond precipitation, soil moisture is a critical regulator of ET, particularly during droughts and their recovery phases. Acting as a buffer, soil moisture tempers ET rates during dry periods and amplifies them after rainfall, as noted in late August. This buffering capacity results in a delay between precipitation events and subsequent ET changes, which is key to understanding drought recovery dynamics. VISEA's data accurately reflect these variations in precipitation, demonstrating its effectiveness in tracking daily ET fluctuations and its reliability for near-real-time monitoring of ET during hydrological extremes.

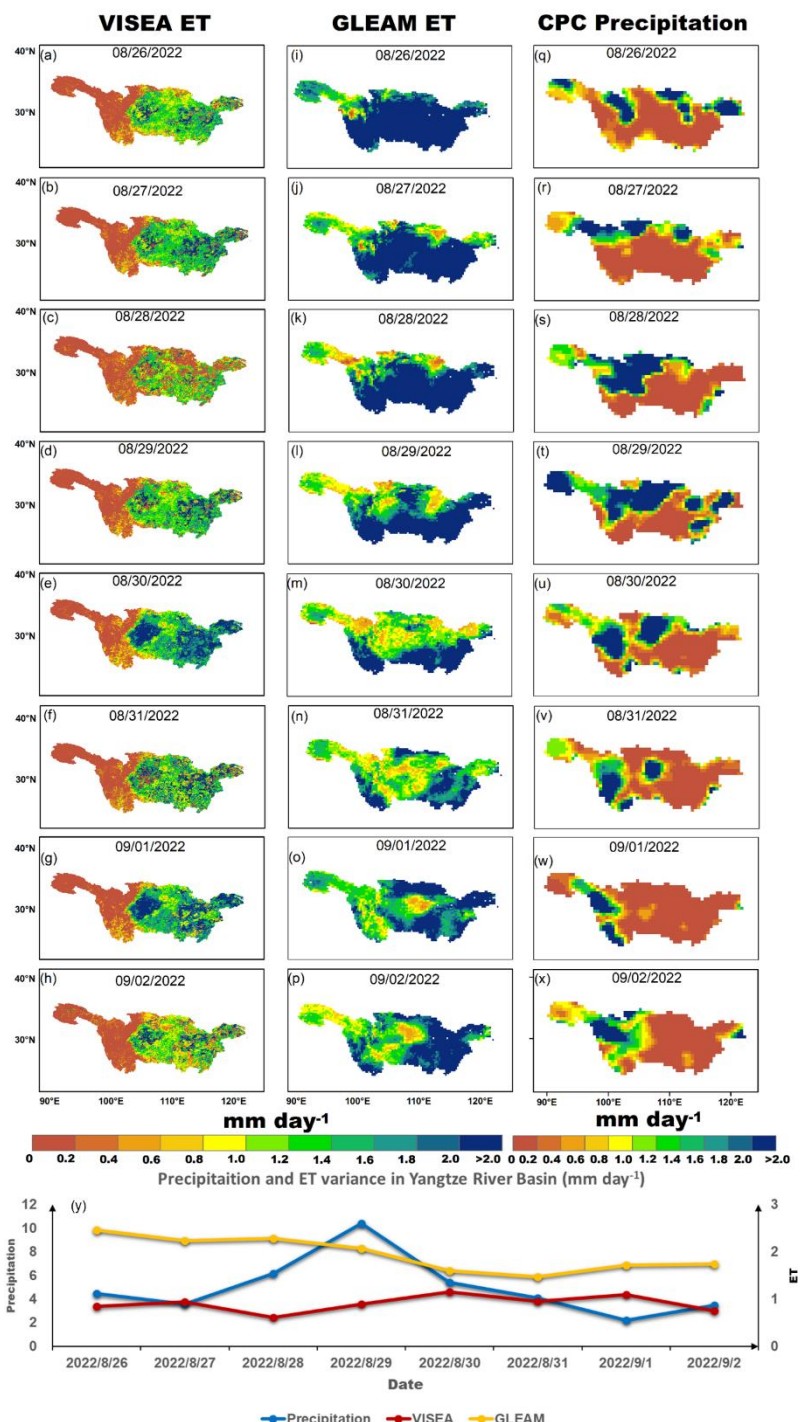

**Figure 9.** Daily ET from VISEA (a-h), GLEAM (i-p), and CPC precipitation (q-x) distributions from August 26th to September 2nd in 2022, alongside daily mean ET and Precipitation variances in the Yangtze River Basin (y) during the same period.

## 5. Discussion

While global ET products (GLEAM, FLUXCOM, AVHRR, MOD 16 and PML ET) require at least 2 weeks to generate global actual ET estimation, we developed VISEA, a satellite-based algorithm which is capable of generating near-real-time evapotranspiration on a daily time step with a resolution of 0.05°.

To assess its accuracy, we compared the calculated ET with data from 149 flux towers around the world
in various land use types.
Scale mismatch is a problem for many satellite-based ET products. The footprints of these flux towers
typically range from 100 to 200 meters, while the VISEA model outputs gridded cells at a resolution of
$0.05° \times 0.05°$ (nearly 25 km²). This discrepancy introduces errors, especially since flux towers require a
uniform fetch, which may not represent the larger gridded cell (Sun et al., 2023). To enhance the validity
of our assessments, we assessed monthly values and spatial patterns of our ET measurements with five
other satellite-based ET products named MOD16, AVHRR, GLEAM, FLUXCOM and PML (Figures 7
and 8).
The VISEA model uses gridded ERA5-Land shortwave downward radiation as its energy input.
Utilizing this input, along with MODIS land surface products, VISEA calculates gridded daily air
temperature and net radiation. These two important intermediate variables are essential for estimating
daily ET. The calculated ET generally matches local measurements and other model-calculated values
well, but we found significant biases (Figures 6 and 7). These biases largely arise from inaccuracies in
the input ERA5-Land shortwave radiation (Figure 3), improper application of the VI-Ts method (Figure
4), and uncertainties in daily net radiation (Figure 5). Next, we look further into the causes of the biases.
Incoming shortwave radiation from ERA5-Land is employed to derive the available energy for
vegetation coverage and bare soil (Eq. 15 and 16), which are the main parameters for calculating daily
ET (Eq. 17). While ERA5-Land is widely utilized as a reanalysis dataset, offering near-real-time land
variables by integrating model data with global observations based on physical laws. However, the
accuracy of shortwave radiation from ERA5-Land seems compromised in savannas (Figure 3) due to the
challenges associated with simulating radiation transmission under land-use changes and aerosol
pollution from natural or anthropogenic sources (Babar et al., 2019; Martens et al., 2020).
Air temperature is an important parameter in determining the daily evaporation fraction of bare soil
(Appendix B), canopy surface resistance, aerodynamic resistance of the bare soil (Appendix C),
atmospheric emissivity (Appendix D),  and available energy for vegetation coverage and bare soil. Since
air temperature is not measured directly by satellites, many other ET products use therefore ground
observations, land models or reanalysis data. In contrast, VISEA derives the air temperature from the
negative linear relationship between vegetation index (VI) and surface temperature (Ts) using the VI-Ts
method (section 2.1.3). It gives very good results under grass land, open shrubland and woody savannas
landcover types, as shown in Figure 4. As previously explained, the VI-Ts method relies on the negative
linear correlation between the Vegetation Index (VI) and surface temperature (Ts) within a $5 \times 5$ grids'
window. Therefore, the variance of VI values across these grid cells and the strength of their negative
correlation are crucial for accurately calculating air temperature (Nishida et al., 2003). However, the VI-
Ts method is less effective in regions like dense forests, bare lands and deserts, where the vegetation
index and temperature data vary little across the $5 \times 5$ grids' window. Also, in regions with freezing
temperatures, the VI-$T_S$ method does not perform well because warmer temperature is related to increased
vegetation, which is the opposite of warmer areas, where there is a positive correlation between the
vegetation index and surface temperature (Cui et al., 2021).
Another bias source of the VISEA model is the uncertainties of daily net radiation, notably
originating from input downward shortwave radiation from ERA5-Land (Figure 2) and VI-Ts estimated
air temperature (Figure 4). The energy budget equation (Eq. 14) and these two figures indicate that net
radiation shows more uncertainties than shortwave radiation and air temperature. At the same time,
assuming a linear relationship between cloud coverage (Eq. 15 and 16) and calculating downward
longwave radiation (Eq. 17 and 18) may be an oversimplification that could introduce uncertainties. Since
available energy for evapotranspiration (ET) depends on net radiation (Eq. 14), addressing these
uncertainties is crucial for enhancing overall model accuracy (Huang et al., 2023b). Future refinements
will contribute to a more precise daily net radiation estimation within the VISEA model.
The VISEA model calculates ET primarily based on vegetation coverage, utilizing it as an indirect
constraint to estimate evapotranspiration. However, this model does not directly incorporate variables
related to water availability, which is a critical factor in ET processes. In tropical regions, where solar
radiation is abundant (available energy), the model tends to overestimate ET due to its emphasis on
vegetation coverage without adequately accounting for the actual water available for evapotranspiration.
This methodology, while effectively capturing the effect of vegetation on ET under varied conditions,
can lead to overestimations in areas where energy availability significantly exceeds water availability,
typical of many tropical regions. Our analysis and subsequent discussion aim to highlight this
characteristic of the VISEA model, acknowledging its implications for ET estimations in such energy-
rich, water-variable environments.
While the VISEA model provides evapotranspiration (ET) globally, its best ET is between 60°N and
90°S, as evidenced by a Nash-Sutcliffe efficiency (NSE) of 0.4 and a correlation coefficient (R) of 0.9 in
Figure 6. VISEA model tends to underestimate ET in colder regions within the 60°N to 90°S latitude
range, such as the western territories of Canada. This underestimation is primarily due to the model's
inability to incorporate evaporation from frozen surfaces into its ET calculations. These discrepancies
arise from several factors: inaccuracies in the ERA5-Land shortwave radiation data (illustrated in Figure
3), the misapplication of the VI-Ts method (explained in Figure 4), and the uncertainties in daily net
radiation (depicted in Figure 5). Designed to amalgamate bare soil and full vegetation coverage, as shown
in Equation 1, the VISEA model encounters difficulties in accurately estimating ET at higher latitudes,
especially in conditions of reduced solar radiation. These challenges are predominantly linked to the
uncertainties associated with ERA5-Land shortwave radiation data, further compounded by increased
cloudiness levels in these regions, as highlighted by Babar et al. (2019). Such uncertainties substantially
impact the model's performance at higher latitudes, affecting its reliability in these conditions.
Nevertheless, VISEA's ET estimates compare favorably with other ET data products in cold regions
above 60°N, as indicated by the latitude zonal mean comparison in Figure 8.
The accuracy of the VISEA model could be enhanced by incorporating additional satellite and
climate data with higher resolution and improved accuracy. Moreover, the delay in providing ET data

could be reduced to three days or less by integrating real-time updated satellite and climate data. We propose developing alternative methods for estimating air temperature and net radiation to enhance accuracy. Additionally, incorporating variables such as soil moisture and water availability into the model could further refine its precision. These improvements provide a roadmap for future research, aiming to significantly enhance satellite-based near-real-time ET modeling.

## 6. Conclusion

Several satellite-based ET products have been developed, but few estimate near-real-time global terrestrial evapotranspiration (ET). We have developed VISEA ET, which only uses satellite-based input data and can provide near-real-time global daily terrestrial ET estimates at a 0.05° spatial resolution. The accuracy of VISEA ET estimates is comparable to existing ET products sooner than existing products. Our evaluations show that VISEA aligns well with measurements from 149 globally distributed tower flux sites on daily and monthly scales. In addition, VISEA captures spatial patterns of evapotranspiration, aligning with GPCC precipitation data across diverse geographical regions, particularly highlighting elevated values in tropical rainforest regions and lower values in arid and semi-arid zones. ET estimates are slightly too high in the Sahara and slightly too low in western Canada. Specifically, daily net radiation and ET estimations of VISEA in Savannah and frozen surfaces need improvements. We plan to address these issues in future developments. The near-real-time global daily terrestrial ET estimates provided by VISEA are valuable for meteorology and hydrology applications, especially for coordinating relief efforts during droughts.

## 7. Code Availability

Python code to synthesise the results and to generate the figures of VISEA results and the codes for generating the global ET products can be obtained through the public repository at https://doi.org/10.6084/m9.figshare.24647721.v1 (Huang, 2023c). The VISEA code for calculating daily ET is written in C and can be executed on Windows 10 using an Intel(R) Core (TM) i7-8565U CPU @ 1.80GHz, 1.99 GHz, 16.0 GB RAM with Visual Studio 2019, or compatible platforms. Additionally, it can run on high-performance computing servers equipped with an Intel(R) Xeon(R) CPU E5-2680 in a CentOS environment. The system is scalable, supporting configurations ranging from 20 nodes and 656 CPUs down to fewer nodes and CPUs as required.

## 8. Data Availability

The VISEA ET data can be obtained from https://doi.org/10.11888/Terre.tpdc.300782 (Huang, 2023a). We are committed to continuously updating this dataset, ensuring that the latest ET data will be consistently and promptly made available.

### 8.1 Input data

MOD11C1 can be obtained at https://e4ftl01.cr.usgs.gov/MOLT/MOD11C1.061/. MOD09CMG
can be obtained at https://e4ftl01.cr.usgs.gov/MOLT/MOD09CMG.061/. MCD43C3 can be obtained at
https://e4ftl01.cr.usgs.gov/MOTA/MCD43C3.061/.     MOD13C1     can     be     obtained     at
https://e4ftl01.cr.usgs.gov/MOLT/MOD13C1.061/.     MCD12C1     can     be     obtained     at
https://e4ftl01.cr.usgs.gov/MOLT/MOD21C1.061/. ERA5-Land shortwave radiation data can be
obtained at https://cds.climate.copernicus.eu/cdsapp#!/dataset/reanalysis-era5-land?tab=form.
**8.2 Evaluation data**
FLUXNET2015 flux towers data (FLUXNET2015: CC-BY-4.0 33) can be obtained at
https://fluxnet.org/data/download-data/. The GLEAM 3.8a ET dataset was obtained from
https://www.gleam.eu/#downloads (an email is required to receive a password for the SFTP). The
FLUXCOM ET dataset was freely available (CC4.0 BY licence) from https://www.fluxcom.org/EF-
Download/ the Data Portal (an email is required to are receive a password for the FTP). MOD16 ET with
the     resolution     of     0.05°     was     freely     downloaded     from
http://files.ntsg.umt.edu/data/NTSG_Products/MOD16/MOD16A2_MONTHLY.MERRA_GMAO_1k
mALB/Previous/.     Additionally,     the     AVHRR     ET     dataset     with     1°     was     sourced     from
http://files.ntsg.umt.edu/data/ET_global_monthly_ORIG/Global_1DegResolution/ASCIIFormat/.
Lastly, the PML ET dataset was obtained from https://www.tpdc.ac.cn/zh-hans/data/48c16a8d-d307-
4973-abab 972e9449627c.
The precipitation from Global Precipitation Climatology Centre (GPCC) data was as obtained at
https://cds.climate.copernicus.eu/cdsapp#!/dataset/insitu-gridded-observations-global-and-
regional?tab=form. The precipitation from Global Unified Gauge-Based Analysis of Daily Precipitation
(CPC) was obtained at https://downloads.psl.noaa.gov/Datasets/cpc_global_precip/precip.2022.nc
Other data that supports the analysis and conclusions of this work is available at
https://figshare.com/articles/dataset/Satellite-based_Near-Real
Time_Global_Daily_Terrestrial_Evapotranspiration_Estimates/24669306 (Huang, 2023d).

## Appendix

**Appendix A. Determining the vegetation fraction calculation:**

$$f_{veg} = \frac{NDVI - NDVI_{min}}{NDVI_{max} - NDVI_{min}} \tag{A1}$$

where the $NDVI$ is the Normalized Difference Vegetation Index and can be calculated as:

$$NDVI = \frac{R_{nir} - R_{red}}{R_{nir} + R_{red}} \tag{A2}$$

where $NDVI_{min}$ is the $NDVI$ of the bare soil without plants and $NDVI_{max}$ is the $NDVI$ of the full vegetation cover, $R_{nir}$ is the near-infrared reflectance and $R_{red}$ is the red reflectance. The daily reflectance $R_{nir}$ and $R_{red}$ were measured by MODIS reflectance data MOD09CMG (Fig. 1). Based on Tang et al. (2009), we set $NDVI_{min} = 0.22$ and $NDVI_{max} = 0.83$. Missing observation for the daily MOD09CMG calculated $NDVI$ data was filled with the 16-day averaged $NDVI$ values in the MOD13Q1data product (Fig. 1).

**Appendix B. Determining of decoupling factor:**

$\Omega_i^*$ is the value of the decoupling factor, $\Omega$, for wet surface. According to Pereira (2004), $\Omega$ and $\Omega^*$ can be expressed as:

$$\Omega = \frac{1}{1+\frac{\gamma}{\Delta+\gamma}\frac{r_c}{r_a}} \tag{B1}$$

$$\Omega^* = \frac{1}{1+\frac{\gamma}{\Delta+\gamma}\frac{r^*}{r_a}} \tag{B2}$$

$$r^* = \frac{(\Delta+\gamma)\rho C_p VPD}{\Delta\gamma(R_n-G)} \tag{B3}$$

where $r_c$ is the surface resistance (s m$^{-1}$); $r_a$ is the aerodynamic resistance (s m$^{-1}$); the calculation details of instantaneous and daily $r_c$ and $r_a$ for vegetation and soil. $r^*$ is the critical surface resistance when the actual evapotranspiration equals the potential evaporation (called equilibrium evapotranspiration, s m$^{-1}$); $\rho$ is the air density (kg m$^{-3}$); $C_p$ is the specific heat of the air (J kg$^{-1}$ K$^{-1}$); $VPD$ is the vapor pressure deficit of the air (Pa). $\Delta$ is the slope of the saturated vapor pressure (Pa K$^{-1}$).

## Appendix C. Determining the resistances of vegetation canopy and bare soil surface

The canopy surface resistance of the vegetation, denoted as $r_{c\,veg}$ (s m$^{-1}$), was determined using the relationship established by Jarvis et al. (1976), is equivalent to:

$$\frac{1}{r_{c\,veg}} = \frac{f_1\,(T_a)f_2\,(PAR)f_3\,(VPD)f_4\,(\varphi)f_5\,(co_2)}{r_{cMIN}} + \frac{1}{r_{cuticle}} \tag{C1}$$

The minimum resistance $r_{cMIN}$ (s m$^{-1}$) is defined as 33 (s m$^{-1}$) for cropland and 50 (s m$^{-1}$) for forest as determined by Tang et al. (2009); the canopy resistance related to diffusion through the cuticle layer of leaves $r_{cuticle}$ is set at 100,000 (s m$^{-1}$) in the Biome-BGC model is according to White et al. (2000). The relationships involving air temperature $T_a$, $f_1(T_a)$ and photosynthetic active radiation $PAR$, $f_2(PAR)$ expressed by the functions provided Jarvis et al. (1976):

$$f_1\,(T_a) = \left(\frac{T_a-T_n}{T_o-T_n}\right)\left(\frac{T_x-T_a}{T_x-T_a}\right)^{\left(\frac{T_x-T_o}{T_o-T_n}\right)} \tag{C2}$$

The minimum, optimal, and maximum temperatures for stomatal activity are denoted as $T_n$, $T_o$ and $T_x$, respectively. As per Tang et al. (2009), $T_n$ is set to 275.85 K, $T_o$ to 304.25 K, and $T_x$ to 318.45 K. The expression for the function $f_2(PAR)$ is provided below:

$$f_2\,(PAR) = \frac{PAR}{PAR+A} \tag{C3}$$

where $PAR$ is photosynthetic active radiation per unit area and time (µ mol m$^{-2}$ s$^{-1}$) calculated by incoming solar radiation multiplied by 2.05 (White et al., 2000); $A$ is a parameter related to photon absorption efficiency at low light intensity, which was set to 152 µ mol m$^{-2}$ s$^{-1}$ 20; Nishida[32] found that in Eq. D1 the following functions can be omitted without great loss of accuracy: the functions depending on vapor pressure deficit, $f_3\,(VPD)$, leaf water potential $f_4\,(\varphi)$ and carbon dioxide vapor pressure, $f_5\,(CO_2)$.

The photosynthetic active radiation per unit area and time ($PAR$), measured in µ mol m$^{-2}$ s$^{-1}$, is computed by multiplying incoming solar radiation by 2.05, as outlined by White et al. (2000). The parameter A, associated with photon absorption efficiency at low light intensity, is established at 152 µ mol m$^{-2}$ s$^{-1}$. Nishida et al. (2003) observed that, in Eq. D1, the functions tied to vapor pressure deficit $f_3\,(VPD)$, leaf water potential $f_4\,(\varphi)$, and carbon dioxide vapor pressure $f_5\,(CO_2)$ can be omitted without significant loss of accuracy. Tang et al. (2009) employed this canopy resistance approach to estimate evapotranspiration (ET) at a 500-meter resolution in the Kalam river basin. The evaluation of their results indicated that the simplification of these calculations did not significantly impact the final accuracy of ET estimates. Additionally, Huang et al. (2017) evaluated this method for 0.05° ET assessments across China. In this study, we follow the methodologies originally developed by Tang et al. (2009) and Nishida (2003), with the goal of enhancing the VISEA model to accurately estimate daily scale evaporation fraction and net radiation. These efforts build on earlier work by Huang et al. (2017, 2021 and 2023b)

that introduced vapor pressure deficit (VPD) and leaf water potential in calculating canopy resistance. However, comparative analyses between VISEA and other models, such as PML and MOD16—particularly PML, which integrates VPD as a limiting factor in estimating GPP and ET—show that VISEA maintains accuracy without significant biases. It is important to note that none of the ET models in our comparison directly incorporate leaf water potential into their canopy resistance calculations. We are committed to addressing these gaps in our future studies.

The aerodynamic resistance of the canopy, denoted as $r_{a\,veg}$ (s m$^{-1}$), is computed for forest cover, grassland, and cropland using the empirical formulae presented by Nishida et al. (2003) for both instantaneous and daily values.

$$\frac{1}{r_{a\,veg\,(forest)}} = 0.008U_{50m} \tag{C4}$$

The wind speed at a height of 50 meters above the canopy ($U_{50m}$) is used to determine the aerodynamic resistance for grassland and cropland, as follows:

$$\frac{1}{r_{a\,veg\,(grassland\,\&\,cropland)}} = 0.003U_{1m} \tag{C5}$$

where $U_{1m}$ is the wind speed 1m above the canopy (m s$^{-1}$). The wind speed as a function of the height z, $U(z)$ can be calculated by the logarithm profile of wind. A recent study found that the velocity log law does not apply to a stratified atmospheric boundary layer (Cheng et al., 2011). Thus D4 and D5 are valid under neutral boundary layer conditions. Since $r_{a\,veg}$ is calculated differently for forests (Eq. D4) and grasslands/croplands (Eq. D5), we used the land cover classes from the yearly International Geosphere-Biosphere Programme (IGBP) (MCD12C1) to identify the land cover and choice the different equation of $r_{a\,veg}$. $U_{50m}$ and $U_{1m}$ were calculated by the logarithm profile of wind:

$$U(z) = U_{shear}\ln\left[\frac{(z-d)}{z_0}\right]/k \tag{C6}$$

where $U_{shear}$ is the shear velocity (m s$^{-1}$); $z$ is the height (m); $d$ is the surface displacement (m); $z_0$ is the roughness length, we followed Nishida et al. (2003), set as 0.005 m for bare soil and 0.01 m for grassland; $k$ is the von Kármán's constant and set as 0.4 following Nishida (Nishida et al., 2003). The shear velocity $U_{shear}$ was calculated as:

$$U_{shear} = U_{1m\,soil}\frac{0.4}{\ln\left(\frac{1}{0.005}\right)} \tag{C7}$$

where the $U_{1m\,soil}$ is the wind speed of bare soil at 1 m height (m s$^{-1}$), it was calculated as:

$$U_{1m\,soil} = 1/0.0015\,r_{a\,soil} \tag{C8}$$

The Vegetation Index-surface Temperature (VI-T$_S$) diagram (Nishida et al., 2003) can be utilized to compute the instantaneous air temperature. This is achieved by utilizing MODIS instantaneous surface temperature/emissivity data (MOD11C1) and daily-calculated NDVI as input parameters.

The aerodynamic resistance of the bare soil, denoted as $r_{a\,soil}$ (s m⁻¹), was determined by Nishida
et al. (2003). This calculation assumes that the maximum surface temperature of bare soil $T_{soil\,max}$ (K)
happens when the sum of latent heat flux and sensible heat flux of the bare soil, referred to as the available
energy of bare soil $Q_{soil}$ (W m⁻²), is utilized as the sensible heat flux, while the latent heat flux is set to
zero.
$$r_{a\,soil} = \frac{\rho C_p (T_{soil\,max} - T_a)}{Q_{soil}}$$     (C9)
$r_{a\,soil}$ is the aerodynamic resistance of the bare soil, (s m⁻¹), $\rho$ is the air density, kg m⁻³; $C_p$ is the
specific heat of the air, (J kg⁻¹ K⁻¹); $T_a$ is the air temperature (K), $Q_{soil}$ is the available energy of bare soil
(W m⁻²).
To compute the canopy surface resistance of bare soil, denoted as $r_{c\,soil}$ (s m⁻¹), we adhere to the
methodologies outlined in the works of Griend and Owe (1994) and Mu et al. (2007):
$$r_{c\,soil} = r_{tot} - r_{a\,soil}$$     (C10)
$$r_{tot} = \frac{1.0}{\left(\frac{T_a}{293.15}\right)^{1.75} \frac{101300}{P}} * 107.0$$     (C11)
The total aerodynamic resistance $r_{tot}$ (s m⁻¹) is composed of the aerodynamic resistance over the
bare soil $r_{a\,soil}$ (s m⁻¹), with atmospheric pressure $P$ set at 101,300 Pa.

**Appendix D. The calculation of atmospheric emissivity for clear sky**
As per Brutsaert (1975), the atmospheric emissivity $\varepsilon_a^d$ for clear sky under standard humidity and
temperature conditions is
$$\varepsilon_a^d = 1.24 \times (e_a^d/T_a^d)^{1/7} \tag{D1}$$
where $e_a^d$ represents the daily water vapor pressure (kPa). To calculated $e_a^d$, it is necessary to
compute the slope of the saturated vapor (Δ) as:
$$\Delta = \frac{4098 \left[0.6108 \exp\left[\frac{17.27 T_a}{(T_a+237.3)}\right]\right]}{(T_a+237.3)^2} \tag{D2}$$
VPD is the vapor pressure deficit of the air (kPa), which is expressed as:
$$\text{VPD} = e^0(T_a) - e_a \tag{D3}$$
$$e^0(T_a) = 0.6108 \exp\left[\frac{17.27 T_a}{(T_a+237.3)}\right] \tag{D4}$$
$$e_a = e^0(T_{dew}) \tag{D5}$$
$$e^0(T_{dew}) = 0.6108 \exp\left[\frac{17.27 T_{dew}}{T_{dew}+237.3}\right] \tag{D6}$$
The expression within parentheses denotes the independent variable, where, $e^0(T_a)$ represents the
saturation vapor pressure (kPa) at the air temperature $T_a$ (°C); $e_a$ is the actual vapor pressure (kPa);
$e^0(T_{dew})$ is the saturation vapor pressure (kPa) at the dew point temperature $T_{dew}$ (°C). For forest, water
surface, and cropland $T_{dew}$ is set to the minimum air temperature during the day. In arid regions such as
bare soil and non-irrigated grassland, $T_{dew}$ may be 2-3 °C lower than $T_{min}$. Therefore, 2 °C is subtracted
is subtracted from $T_{min}$ in arid and semiarid areas to derive $T_{dew}$. While these simplifications might
introduce a bias in the final calculated ET value, our initial results indicate that the effect is negligible.

**Acknowledgements**

This study is supported by the National Key Research and Development Program of China (No.2017YFA0603703). We employed ChatGPT3.5 to enhance the quality of our English writing and grammar.

**Author contributions**

L. H. had the original idea and drafted the paper with help from Y. L.; J. M. C. Q. T., T. S., W. C. and W. S. participated in the discussion and the many manuscript revisions.

**Competing interests**

The authors declare no competing interests.

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
