# Peer review of "Satellite-based Near-Real-Time Global Daily Terrestrial Evapotranspiration Estimates"

_Earth System Science Data, 2023_

## Author Comment (AC1)

**Response to Liu Mingliang**

essd-2023-495

Title: Satellite-based Near-Real-Time Global Daily Terrestrial Evapotranspiration Estimates

Author(s): Lei Huang et al.

MS type: Data description paper

In the reply, the reviewers' comments are in *italics*, our response is in normal text, and quotes from the manuscript are in **blue**.
* * *
*This paper claimed that the advantage of this new global ET data product is that it is near-real-time which has about one week's delay, but with a little bit better or comparable accuracies with other data products. The authors should discuss how much contribution of this one-week earlier (than other models/data products) to the community and how possible accuracies could be improved if more observations such as climate data are added and the estimation be delayed to two-weeks or even further. The authors also need make the near-real-time data public available or accessible (i.e. the data products will be updated at real time if being needed); or the source code or tool could be used to calculate global ET through cloud computing platforms, such as GEE. The bias on estimated air temperature and net radiation (might also need add some comparisons with other regional and global data sets on net radiation estimations) should be addressed before publishing this data products since the estimated air temperature and the application of shortwave radiation as input is the bases of this near-real-time ET products. What are the uncertainties coming from the estimated EF by using calculated vegetation and soil resistance, rather than directly from remote sensed information, such as surface temperature and NDVI (through Ts-VI triangle method)?*

Re: We thank the reviewers! We have added discussion about contribution of the one-week earlier to the community and how possible accuracies could be improved at Lines 702-707:

"The VISEA ET product provides near-real-time global evapotranspiration (ET) data with a mere one-week delay and a daily resolution of 0.05 degrees. It empowers researchers by providing access to information on land surface water consumption in near-real-time, which is crucial for monitoring and predicting droughts, and enables decision-makers to make well-informed choices. This not only enhances research efficiency but also supports more effective and expedited actions within the scientific and environmental research community. The accuracy of the VISEA model could be enhanced by incorporating additional

satellite and climate data with higher time resolution. Moreover, the one-week delay in providing ET data could be reduced to three days or less by integrating real-time updated satellite observation data."

We have uploaded our ET data at National Tibetan Plateau Data Center Third Pole Environment Data Center at lines 746-747:

[revised manuscript text omitted]

*What is the differences of this data product with other global ET on the long-term trend, inter-annual variation, and under extreme climate events?*

Re: We have added long-term trends and inter-annual variations for these ET products in Figure 8 and we added the description about the long-term trend, inter annual variation at Lines 554-556:
"The VISEA ET product demonstrates consistent spatial distribution patterns among the six ET products across various years, both in terms of annual means (a-g) and latitude zonal means (h). These patterns align closely with the precipitation distribution data from GPCC."

And Lines 583-588 "Regarding the inter-annual monthly variations, panel (i) shows the fluctuations in ET across different years for the analyzed ET products and precipitation data. The graph reveals a rhythmic pattern of ET across the years, VISEA with other ET products showed distinctive peaks and troughs that correspond to seasonal changes and inter-annual climate variability. The ET products' data exhibit a close

alignment with the precipitation patterns reported by GPCC, highlighting the interconnectedness between ET and precipitation as climatic variables."

[Figure]

Figure 8. The spatial distribution of the multi-year average (a-g), the zonal mean (h) and inter-annual variation (i) of (a) GPCC (2001-2019), (b) VISEA (2001-2020), (c) GLEAM (2001-2020), (d) FLUXCOM (2001-2016), (e) AVHRR (2001-2006), (f) MOD16 (2001-2014) and (g) PML (2003-2018).

To demonstrate the performance of our data product under extreme climate conditions, we have added the 5th and 95th percentage spatial patterns in Supplementary Figure S1 and Figure S2. At lines 586-588: "It also exhibits similar distributions to other ET products, both below the 5th percentile (Figure S4) and above the 95th percentile (Figure S5)."

[Figure]

**Figure S4.** Monthly mean precipitation (a) and ET (b-g) when values less than 5th percentile (mm month-1) (a) GPCC (2001-2019), (b) VISEA (2001-2020), (c) GLEAM (2001-2020), (d) FLUXCOM (2001-2016), (e) AVHRR (2001-2006), (f) MOD16 (2001-2014) and (g) PML (2003-2018).

[Figure]

**Figure S5.** Monthly mean precipitation (a) and ET (b-g) when values large than 95$^{th}$ percentile (mm month$^{-1}$) (a) GPCC (2001-2019), (b) VISEA (2001-2020), (c) GLEAM (2001-2020), (d) FLUXCOM (2001-2016), (e) AVHRR (2001-2006), (f) MOD16 (2001-2014) and (g) PML (2003-2018).

*Line 228: correct the label for surface temperature.*

Re: We have corrected the label for surface temperature at Lines 226:

**"2.1.3 The calculation of daily air temperature, $T_a^d$ and surface temperature, $T_s^d$"**

*L118-120: this description seems not correct since VISEA also use the thermal information such as surface temperature.*

Re: we modified this sentence at line 143 – 145:

"Unlike energy budget-based ET algorithms (such as SEBS, METRIC, and Alexi), which calculate ET (latent heat flux) as the residual of the net radiation, subtracting soil heat flux and sensible heat flux."

*L209-216: how Ta is estimated for each 0.05 degree pixel?*

Re: we explain the calculation progress of air temperature at lines 231-237:

"This method was developed based on the empirical linear relationship between surface temperature (Ts) and Vegetation Index (VI). Surface temperature increases when the vegetation index decreases, and conversely, surface temperature decreases when the vegetation index increases. By defining a "window" formed by the neighboring 5 * 5 grid cells, the scatter plot of these 25 grid cells' VI and Ts typically exhibits a triangular (or trapezoidal) distribution. In this scatter plot, we identify the "warm edge" (characterized by a low vegetation cover fraction and high Ts) and the "cold edge" (marked by a high vegetation cover fraction and low Ts).

Through simple interpolation, Ts corresponding to any given vegetation condition within the range of the "warm edge" and "cold edge" can be determined. The lowest Ts could be determined by the highest VI,

and the highest Ts could be determined by the lowest VI. Therefore, following Nishida et al. (2003), under the assumption that the lowest surface temperature equals the air temperature (Ta), we can derive the daily air temperature."

*L238-246: how is air temperature of each pixel estimated?*

Re: we explained the calculation progress at lines 231-237 above.

*L297-301: rephrase.*

Re: we have rephrased this sentence at lines 308-310:
"We evaluated the accuracy of the input ERA5-Land shortwave radiation, estimated daily net radiation, air temperature, and ET by comparing them against measurements from FLUXNET2015 (Pastorello et al., 2020). The data from FLUXNET2015 can be obtained at https://fluxnet.org/data/download-data."

*L390-399: confusing on the differences between Ts and Ta.*

Re: we explained at line 168: "*Ts*, land surface temperature" and at lines 32: "…air temperature (Ta)."

Figs 3-6: explain what frequency mean.

Re: we added the explanation of "frequency" at Figure 3- Figure 6:

[Figure]

**Figure 3.** The scatter plot of downward solar radiation from ERA5-Land (ERA5_Rd) compared with local instruments measurements (Obv_Rd) under 12 IGBP land cover types: CRO (Croplands), CSH (Closed shrublands), DBF (Deciduous broadleaf forests), DNF (Deciduous needle leaf forests), EBF (Evergreen broadleaf forests), ENF (Evergreen needle leaf forests), GRA (Grasslands), MF (Mixed forests), OSH (Open shrublands), SAV (Savannas), WSA (Woody savannas), WET (Permanent wetlands). The red dotted line is the 1:1 line. N is the number of data points, NSE is Nash-Sutcliffe Efficiency, R is correlation coefficients, RMSE is Root Mean Square Error, RMSEs is systematic RMSE, and RMSEu is unsystematic RMSE. The Frequency denotes the probability density estimated through the KDE method with a Gaussian kernel, and it is then scaled to ensure that the maximum value of the probability density function equals 1. P is the P-Value for the Correlation Coefficient.

[Figure]

**Figure 4.** The scatter plot of daily air temperature simulated by VISEA (VISEA_Ta) compared with local instruments measurements (Obv_Ta) under 12 IGBP land cover types: CRO (Croplands), CSH (Closed shrublands), DBF (Deciduous broadleaf forests), DNF (Deciduous needle leaf forests), EBF (Evergreen broadleaf forests), ENF (Evergreen needle leaf forests), GRA (Grasslands), MF (Mixed forests), OSH (Open shrublands), SAV (Savannas), WSA (Woody savannas), WET (Permanent wetlands). The red dotted line is the 1:1 line. N is the number of data points, NSE is Nash-Sutcliffe Efficiency, R is correlation coefficients, RMSE is Root Mean Square Error, RMSEs is systematic RMSE, and RMSEu is unsystematic RMSE. The frequency denotes the probability density estimated through the Kernel Density Estimation, KDE method with a Gaussian kernel, and it is then scaled to ensure that the maximum value of the probability density function equals 1.

[Figure]

**Figure 5.** The scatter plot of daily net radiation simulated by VISEA (VISEA_Rn) compared with local instruments measurements (Obv_Rn) under 12 IGBP land cover types: CRO (Croplands), CSH (Closed shrublands), DBF (Deciduous broadleaf forests), DNF (Deciduous needle leaf forests), EBF (Evergreen broadleaf forests), ENF (Evergreen needle leaf forests), GRA (Grasslands), MF (Mixed forests), OSH (Open shrublands), SAV (Savannas), WSA (Woody savannas), WET (Permanent wetlands). The red dotted line is the 1:1 line. N is the number of data points, NSE is Nash-Sutcliffe Efficiency, R is correlation coefficients, RMSE is Root Mean Square Error, RMSEs is systematic RMSE, and RMSEu is unsystematic RMSE. The frequency denotes the probability density estimated through the Kernel Density Estimation, KDE method with a Gaussian kernel, and it is then scaled to ensure that the maximum value of the probability density function equals 1.

[Figure]

**Figure 6.** The scatter plot of daily ET simulated by VISEA (VISEA_ET) compared with local instruments measurements (Obv_ET) under 12 IGBP land cover types: CRO (Croplands), CSH (Closed shrublands), DBF (Deciduous broadleaf forests), DNF (Deciduous needle leaf forests), EBF (Evergreen broadleaf forests), ENF (Evergreen needle leaf forests), GRA (Grasslands), MF (Mixed forests), OSH (Open shrublands), SAV (Savannas), WSA (Woody savannas), WET (Permanent wetlands). The red dotted line is the 1:1 line. N is the number of data points, NSE is Nash-Sutcliffe Efficiency, R is correlation coefficients, RMSE is Root Mean Square Error, RMSEs is systematic RMSE, and RMSEu is unsystematic RMSE. The frequency denotes the probability density estimated through the Kernel Density Estimation, KDE method with a Gaussian kernel, and it is then scaled to ensure that the maximum value of the probability density function equals 1.

*Figure 8: suggest adding a plot to show the global annual ET during the study period from these various data products.*

Re: We have added a plot to show the global annual ET during the study period at Figure 8 before.

*L515-529: need more precised description. Is there no regions with energy limited ET?  Should be better to describe the differences in regions with moisture and energy limited ET.*

Re: we added more precise description at lines 559-566:
"The available water for evaporation and transpiration is abundant, and the primary constraint on evapotranspiration lies in the availability of energy to drive the process. In such conditions, water availability is not a limiting factor, allowing for ample potential evapotranspiration."

We also added more precise description at lines 573-575:
"In these areas, there is a surplus of available energy, and the primary limitation on ET stems from the availability of water. This implies a high atmospheric water demand, often quantified as potential evapotranspiration (potential ET)."

We added the describe the differences in regions with moisture and energy limited ET at lines 576-582:
"In regions with moisture-limited evapotranspiration (ET), the primary constraint on ET arises from the limited availability of water. These areas typically experience insufficient precipitation or water supply, leading to a situation where the atmospheric demand for moisture exceeds the available water resources. On the other hand, regions with energy-limited ET face limitations due to inadequate energy for the process of evaporation and transpiration. This can be influenced by factors such as cloud cover, shading, or other conditions that limit the absorption of solar radiation. In such areas, even if there is an ample water supply, the lack of sufficient energy hinders the rate of evapotranspiration."

L530-546: can other global data product reveal the same pattern? Also need adding the conditions before 8/27/2022 to support the point that estimated ET could represent the soil moisture condition. Need clearly highlight the points for introducing this case analysis.

Re: We have incorporated daily scale GLEAM ET and CPC precipitation data from August 26th, 2022, which precedes the specified date of August 28, 2022. But we found GLEAM failed to capture the variability of ET during this drought and exhibited a negative correlation with precipitation data from CPC, so we wouldn't discuss it further in this context. And we added more description of Figure 9 at lines 602-612:

"VISEA ET graphically illustrates the evolving drought conditions: with notably low ET levels (below 1 mm day$^{-1}$) across the basin on August 26th to 28th, evidenced in panel (a-c). A notable increase in precipitation on August 29th, reflected in panels (s) and (u), correlates with an upswing in ET values (surpassing 1 mm day$^{-1}$) throughout the basin, as visualized in panels (d-f). The graph in panel (y) displays the variances in mean ET and precipitation within the basin over this timeframe, highlighting a significant rise in ET (up to 11 mm day$^{-1}$) on August 30$^{th}$, which corresponds with the observed increase precipitation (reaching 11 mm day$^{-1}$) on August 29$^{th}$.

VISEA's ET data align closely with the variances observed in the CPC precipitation data, showcasing its effectiveness in capturing daily ET fluctuations, especially during and after the drought conditions. It accurately reflects the dip and subsequent recovery in ET values following the precipitation events, indicating its robustness in near-real-time monitoring of ET during such hydrological extremes."

[Figure]

Figure 9. Daily ET from VISEA (a-h), GLEAM (i-p), and Precipitation (q-x) distributions from August 25th to September 2nd in 2022, alongside daily mean ET and Precipitation variances in the Yangtze River Basin (y) during the same period.

*L579-582: It is confusing.*

Re: we explained at lines 650-653:

"As previously explained, the VI-Ts method relies on the negative linear correlation between the Vegetation Index (VI) and surface temperature (Ts) within a 5 × 5 grid. Therefore, both the variance of VI values across these grid cells and the negative correlation are essential for calculating the air temperature."

*L584: are you talking about the estimated air temperature?*

Re: yes, in this paragraph, we are talking about the uncertainties of using VI-T$_S$ method to calculate air temperature.

*L586-587: should be more specific. What does "VISEA relies solely on vegetation coverage as an indirect constraint" mean?*

Re: we added more specific description at lines 658-666:

"Another bias source of the VISEA model is the uncertainties of daily net radiation, notably originating from input downward shortwave radiation from ERA5-Land (Figure 2) and VI-Ts estimated air temperature (Figure 4). The energy budget equation (Eq. 11) and these two figures indicate that net radiation shows more uncertainties than shortwave radiation and air temperature. At the same time, assuming a linear relationship between cloud coverage (Eq. 12 and 13) and the calculation of downwards longwave radiation (Eq. 14 and 15) may be an oversimplification that could introduce uncertainties. Since available energy for evapotranspiration (ET) depends on net radiation (Eq. 16 and 17), addressing these uncertainties is crucial for enhancing overall model accuracy (Brutsaert, 1975; Huang et al., 2023). Future refinements will contribute to a more precise daily net radiation estimation within the VISEA model."

*L604-605: where this conclusion come from?*

Re: We modified this conclusion at lines 727-728:

"It demonstrates competitive correlation coefficients and Nash-Sutcliffe efficiencies (NSEs) across most land cover types but exhibits higher biases."

*L607-608: the claim that "VISEA aligns with GPCC a... in most areas worldwide" is confusing. Does VISEA generate precipitation?*

Re: we revised this sentence at lines 730-734:

"VISEA consistently demonstrates spatial patterns aligned with GPCC in most areas, featuring elevated values in tropical rainforest regions and lower values in arid and semi-arid zones. This alignment underscores VISEA's proficiency in portraying the spatial distribution of evapotranspiration, offering valuable insights into water consumption dynamics across diverse geographical regions."

L668-670: add more literature to support this claim. Why are the effects of VPD and leaf water potential on canopy surface resistance so minor that it could be removed totally in the equation?

Re: we have added more literature to support this claim at Lines 845-850:

"Tang et al. (2009) employed this canopy resistance approach to estimate evapotranspiration (ET) at a 500 meter resolution in the Kalam river basin. The evaluation of their results indicated that the simplification of these calculations did not significantly impact the final accuracy of ET estimates. Additionally, Huang et al. (2017, 2021, and 2023) evaluated this method for 0.05 degree ET assessments across China. The evaluation results also demonstrated that the reduction in vapor pressure deficit (VPD) and leaf water potential had minimal effects on the final ET estimates."

---

## Author Comment (AC2)

**Response to Seungcheol Oh**

essd-2023-495

Title: Satellite-based Near-Real-Time Global Daily Terrestrial Evapotranspiration Estimates

Author(s): Lei Huang et al.

MS type: Data description paper

Iteration: Minor revision

In the reply, the reviewers' comments are in *italics*, our response is in normal text, and quotes from the manuscript are in **blue**.
* * *
*Summary:*

*The study introduces a novel approach leveraging the Moderate Resolution Imaging Spectroradiometer (MODIS) to deliver global daily actual evapotranspiration (ET) estimates with a spatial resolution of 0.05°, available within a week of satellite measurements. VISEA employs a combination of a vegetation index-temperature triangle method, a daily evaporation fraction method, and a net radiation calculation that incorporates cloud coverage, utilizing inputs from both ERA5-Land and MODIS land products. The algorithm's efficacy is validated through comparisons with data from 149 flux towers, other satellite-based ET products, and GPCC precipitation data, demonstrating VISEA's comparable performance.*

*General Comments:*

*The manuscript is promising and contributes valuable insights to the field of Earth System Science Data. However, it requires minor revisions before it can be considered for publication. My suggestions mainly pertain to enhancements in figures and tables, as well as a need for a more in-depth discussion.*

Re: We thank the reviewers! We have modified the manuscript following your comments from reviewers. Please see further responses below.

*Abstract: Line 39: Delete ')'.*

Re: we have deleted ')' in the Abstract at lines 38-39:

…e.g., surface reflectance, land surface temperature/emissivity, land cover products, vegetation indices, and albedo as inputs.

*Introduction:*

*Line 69-87: Are there any other satellite-based daily ET products not covered in this part? If not, it's recommended to more clearly highlight why these specific ET products were chosen for discussion. Clarify that each represents unique algorithmic approaches and are widely recognized within the scientific community for their contributions to global ET estimation.*

Re: The reasons for choosing these products have been clarified. We have revised this paragraph at lines 77-79:

"The selected ET products discussed in this study embody diverse and innovative algorithmic approaches that have significantly contributed to global ET estimation and gained recognition within the scientific community."

*Table 2:*

*Please double-check the time periods listed for MOD16 and GLEAM in Table 2. It's possible that more recent data are available for both datasets, extending beyond the years currently noted in your table. Additionally, there seems to be a discrepancy between the time period for MOD16 mentioned in Figure 8, which is listed as 2001-2014, and what's noted in Table 2 as 2001-2013.*

Re: we have double-checked the availability and coverage periods of MOD16 and GLEAM. We confirm that the MOD16 data, with a spatial resolution of 0.05°, is indeed available for the period from 2001 to 2014. We have updated Table 2 to accurately reflect this time period. Furthermore, we have verified that the latest available data from the GLEAM dataset extends up to the year 2020 and we used FLUXCOM to replace the old GBAF data. These updated dataset have been utilized in our analysis and is correctly cited in both Figure 7, Figure 8 and Figure 9 for the evaluation.

**Table 2.** The five global girded ET products and one precipitation product used for comparison with our near-real-time global daily terrestrial ET estimates.

| Product name | Spatial/Temporal resolution | Time period | Theory |
| --- | --- | --- | --- |
| GLEAM | 0.25°/Monthly | 2001-2022 | Priestly-Taylor Equation |
| FLUXCOM | 0.5°/Monthly | 2001-2016 | Machine learning |
| MOD16 | 0.05°/Monthly | 2001-2014 | Penman-Monteith Equation |
| AVHRR | 1°/Monthly | 2001-2006 | Improved Penman-Monteith Equation |
| PML | 0.05°/8-day | 2003-2018 | Penman-Monteith Equation and a diagnostic biophysical model |
| GPCC | 0.25°/Monthly | 2001-2019 | in-situ observations |
| GPC | 0.5°/Daily | 08/28/2022-09/01/2022 | Global Unified Gauge-Based Analysis of Daily Precipitation |

*I suggest adding a comparison of the annual variation of ET with latitude for different remote sensing products to Figure 8 for an improved analysis. For reference, please see Figure 3 in the paper by Chen et al., 2021. https://doi.org/10.1029/2020JD032873*

Re: We have added the comparison of the annual variation of ET with latitude for different remote sensing products to Figure 8 and add the paper as reference (Chen et al., 2021). And we added the full citation.

Chen, X., Su, Z., Ma, Y., Trigo, I., and Gentine, P.: Remote Sensing of Global Daily Evapotranspiration based on a Surface Energy Balance Method and Reanalysis Data, Journal of Geophysical Research: Atmospheres, 126, e2020JD032873, https://doi.org/10.1029/2020JD032873, 2021.

[Figure]

Figure 8. The spatial distribution of the multi-year average (a-g), the zonal mean (h) and inter-annual variation (i) of (a) GPCC (2001-2019), (b) VISEA (2001-2020), (c) GLEAM (2001-2020), (d) FLUXCOM (2001-2016), (e) AVHRR (2001-2006), (f) MOD16 (2001-2014) and (g) PML (2003-2018).

*Discussion:*

*1.     The discussion could benefit from a more detailed analysis of the methodological uncertainties inherent in VISEA. Consider exploring not only the input data challenges but also the underlying assumptions and limitations of the model itself.*

Re: we added the discussion of the VI-Ts method uncertainties in VISEA at lines 648-655:

"As previously explained, the VI-Ts method relies on the negative linear correlation between the Vegetation Index (VI) and surface temperature (Ts) within a 5 × 5 grid. Therefore, both the variance of VI values across these grid cells and the negative correlation are essential for calculating the air temperature. However, in regions where the vegetation index and temperature data in adjacent grid cells show small variations, such as dense forests and bare lands and deserts. Also, in regions with freezing temperatures, the VI-TS method does perform well, because warmer temperature is related to increased vegetation, opposite the other regions, where there is a negative."

*2.     The discussion mentions several critical points regarding the sources of bias and inaccuracies but seems to lack sufficient citation from existing literature to contextualize these findings within the broader field.*

Re: we added the references at line 637-640:

"However, the accuracy of shortwave radiation from ERA5-Land seems compromised in savannas (Figure 3) due to the challenges associated with simulating radiation transmission under land-use changes and aerosol pollution from natural or anthropogenic sources (Babar et al., 2019; Martens et al., 2020)."

at lines 653-655,

"Also, in regions with freezing temperatures, the VI-TS method does perform well, because warmer temperature is related to increased vegetation, opposite the other regions, where there is a positive correlation between the vegetation index and surface temperature (Cui et al., 2021)."

and lines 661-664

"Since available energy for evapotranspiration (ET) depends on net radiation (Eq. 20 and 21), addressing these uncertainties is crucial for enhancing overall model accuracy (Brutsaert, 1975; Huang et al., 2023)."

*3.      Conclude the discussion with specific suggestions for future research directions that could address the identified gaps and uncertainties. This may include the development of alternative methods for estimating air temperature and net radiation, the incorporation of additional variables such as soil moisture and water availability into the model, or the potential for integrating machine learning techniques to improve estimation accuracy.*

Re: we added the discussion with specific suggestions for future research directions at lines 706-715:

"The accuracy of the VISEA model could be enhanced by incorporating additional satellite and climate data with higher resolution and improved accuracy. Moreover, the delay in providing ET data could be reduced to three days or less by integrating real-time updated satellite and climate data. In response to the suggestion to conclude our discussion with specific recommendations for future research directions, we recognize the importance of addressing the identified gaps and uncertainties. We propose exploring the development of alternative methods for estimating air temperature and net radiation to provide more accurate and reliable models. Additionally, incorporating variables such as soil moisture and water availability into the model could further refine its precision. By integrating these suggestions, we aim to outline a comprehensive roadmap for future research that builds upon our findings, significantly contributing to the enhancement of environmental modeling and prediction within the field."

---

## Author Comment (AC3)

**Response to Ren Wang**

essd-2023-495

Title: Satellite-based Near-Real-Time Global Daily Terrestrial Evapotranspiration Estimates

Author(s): Lei Huang et al.

MS type: Data description paper

In the reply, the reviewers' comments are in *italics*, our response is in normal text, and quotes from the manuscript are in **blue**.
* * *
*Comments to Huang et al. Satellite-based Near-Real-Time Global Daily Terrestrial Evapotranspiration Estimates*

*This manuscript aims to present a near-real-time global terrestrial evapotranspiration (ET) estimate by utilizing their previous developed VISEA algorithm, satellite observation, and ERA5-land reanalysis data. The development of a near-real-time ET product holds great significance for drought monitoring, water resources management, and climate change study. I am pleased to witness this progress, and I think the manuscript may have potential to be published in ESSD. However, there are several major concerns that the authors should further revise or clarify.*

*Major comments:*

*1.       The authors essentially use an energy balance based approach to estimate ET, but they do not discuss the limitations of this methodology. In my experience, the EF/energy balance method may not perform well in the local winter season or at high latitudes (e.g., 60-90N) due to energy Although the authors only utilize ERA5-Land downward shortwave radiation values greater than 0, they do not discuss the difficulties and limitations of the methods. However, since the focus of the study in on daily-scale ET estimation, this issue remains relevant and requires further attention. If I am right, the resulting ET estimates in this study is a near-global ET product. In the northern hemisphere (60-90 N), the ET estimates may be unavailable or miss data for nearly half of the year.*

Re: We acknowledge these limitations and have discussed their potential impact on our study results, and emphasized the uncertainties in evapotranspiration estimates at latitudes between 60°N and 90°N, especially during winter when surface evaporation from frozen surfaces is not adequately represented, at lines 677-688:

"Furthermore, the VISEA model exhibits a tendency to underestimate ET in colder regions within the 60°N to 90°N latitude range, such as the western territories of Canada. This underestimation is primarily due to the model's inability to incorporate evaporation from frozen surfaces into its ET calculations. These discrepancies arise from several factors: inaccuracies in the ERA5-Land shortwave radiation data (illustrated in Figure 3), the misapplication of the VI-Ts method (explained in Figure 4), and the uncertainties in daily net radiation (depicted in Figure 5). Designed to amalgamate bare soil and full vegetation coverage as depicted in Equation 1, the VISEA model encounters difficulties in accurately estimating ET at higher latitudes, especially in conditions of reduced solar radiation. These challenges are predominantly linked to the uncertainties associated with ERA5-Land shortwave radiation data, further compounded by increased cloudiness levels in these regions, as highlighted by Babar et al. (2019). Such uncertainties have a substantial impact on the model's performance at higher latitudes, affecting its reliability in these conditions."

Despite these challenges, our analysis confirms the VISEA model's ability to provide valuable ET estimates during the growing season, evidenced by a high Nash-Sutcliffe efficiency (NSE) of 0.4 and a correlation coefficient (R) of 0.9 when compared against local measurements. These findings support the model's applicability for ET estimation in the 60°N to 90°N latitude range, highlighting its effectiveness and relevance during the vegetative growth period."

*2.      The validation of the new product appears to be insufficient. I recommend adding regional average curves to compare their changes over time. Additionally, while Figure 8 presents the spatial distribution characteristics of the multi-year average, what about the extreme values (e.g., 5th and 95th percentiles)? Considering these products are averaged over different time periods, does it affects the comparison results? Clarifying these aspects will enhance the robustness of the validation.*

Re: We have added the regional average curves to compare their changes *over time* at Figure 8 (i) and added the 5th and 95th percentiles of the ET data at lines 556-558:
"It also exhibits similar distributions to other ET products, both below the 5$^{th}$ percentile (Figure S4) and above the 95$^{th}$ percentile (Figure S5). "
And at lines 583-590:

" Regarding the inter-annual monthly variations, panel (i) shows the fluctuations in ET across different years for the analyzed ET products and precipitation data. The graph reveals a rhythmic pattern of ET across the years, VISEA with other ET products showed distinctive peaks and troughs that correspond to seasonal changes and inter-annual climate variability. The ET products' data exhibit a close alignment with the precipitation patterns reported by GPCC, highlighting the interconnectedness between ET and precipitation as climatic variables. Notably, FLUXCOM consistently presents higher ET estimations compared to the other products, and GLEAM's ET estimations are also slightly higher during the winter, indicating a trend of systematic overestimation in these products relative to the others in the dataset."

[Figure]

Figure 8. The spatial distribution of the multi-year average (a-g), the zonal mean (h) and inter-annual variation (i) of (a) GPCC (2001-2019), (b) VISEA (2001-2020), (c) GLEAM (2001-2020), (d) FLUXCOM (2001-2016), (e) AVHRR (2001-2006), (f) MOD16 (2001-2014) and (g) PML (2003-2018).

[Figure]

**Figure S4.** Monthly mean precipitation (a) and ET (b-g) when values less than 5th percentile (mm month-1)
(a) GPCC (2001-2019), (b) VISEA (2001-2020), (c) GLEAM (2001-2020), (d) FLUXCOM (2001-2016), (e)
AVHRR (2001-2006), (f) MOD16 (2001-2014) and (g) PML (2003-2018).

[Figure]

**Figure S5.** Monthly mean precipitation (a) and ET (b-g) when values large than 95th percentile (mm month-1) (a) GPCC (2001-2019), (b) VISEA (2001-2020), (c) GLEAM (2001-2020), (d) FLUXCOM (2001-2016), (e) AVHRR (2001-2006), (f) MOD16 (2001-2014) and (g) PML (2003-2018).

3.      The objective of this study is to provide a near-real-time global ET product, yet the methodology duplicates information from the authors' previously published paper (Huang et al., 2021, Earth and Space Science) (e.g., Figure1, the description of the VISEA model, and the decoupling parameter for daily EF). Therefore, it is very crucial to carefully address and further clarify the distinctions between these repeated details.

Re: we added the explanations of the differences between the algorithm from 2021 at lines 197-200:

"…different from the former study provided by Huang et al., (2023), which set we $\varepsilon_s^d$ and $\varepsilon_a^d$ equal, we calculated the $\varepsilon_a^d$ by Appendix B flowing study of Brutsaert, (1975) and Wang and Dickinson(2013), $\varepsilon_s^d$ can be retried by MOD11C1; $\sigma$ is the Stefan-Boltzmann constant; $T_a^d$ is the daily near surface air temperature (K); $T_s^d$ is the daily surface temperature (K). We account for the influence of clouds by assuming a linear correlation between downward longwave radiation and cloud coverage in the calculation of downwards longwave radiation based on the study of Huang et al., (2021)."

*Minor comments:*

1.      The near-real-time ET proposed in this study primarily relies on MODIS Land product at 05 degrees and ERA5 data at 0.1 degrees. Shouldn't the ET products should be limited to a relatively coarse resolution of 0.1 degree?

Re: we have included a discussion in our manuscript at Lines 401-408:

"Our local scale evaluation, as demonstrated in Figure 3, supports our stance that this resolution disparity between MODIS Land product at 05 degrees and ERA5 data at 0.1 degrees minimally impacts the final ET product's accuracy. This approach is consistent with the methodologies adopted in the studies by Huang et al. (2017, 2021, 2023), which effectively utilized MODIS land products at a 0.05-degree resolution in conjunction with downward shortwave radiation data at a 0.1-degree resolution from the China Meteorology Forcing Dataset. Such precedents underscore the feasibility of integrating these resolutions for ET estimation, bolstering our confidence in the methodological integrity of our study despite the noted resolution differences."

2.       Regarding Figure 3, it would be beneficial to include the level of significance of the correlation analysis.

Re: We added the significance of the correlation in Figure 3. Specifically, we annotated the figure to indicate that all Rd values derived from ERA5 exhibit very low P-values (<0.01). At lines 387-388:

"This indicates a statistically significant correlation between the input shortwave radiation from ERA5 and the local measurements."

[Figure]

**Figure 3.** The scatter plot of downward solar radiation from ERA5-Land (ERA5_Rd) compared with local instruments measurements (Obv_Rd) under 12 IGBP land cover types: CRO (Croplands), CSH (Closed

shrublands), DBF (Deciduous broadleaf forests), DNF (Deciduous needle leaf forests), EBF (Evergreen broadleaf forests), ENF (Evergreen needle leaf forests), GRA (Grasslands), MF (Mixed forests), OSH (Open shrublands), SAV (Savannas), WSA (Woody savannas), WET (Permanent wetlands). The red dotted line is the 1:1 line. N is the number of data points, NSE is Nash-Sutcliffe Efficiency, R is correlation coefficients, RMSE is Root Mean Square Error, RMSEs is systematic RMSE, and RMSEu is unsystematic RMSE. The Frequency denotes the probability density estimated through the KDE method with a Gaussian kernel, and it is then scaled to ensure that the maximum value of the probability density function equals 1. P is the P-Value for the Correlation Coefficient.

*3.        Table 3: Why does the VISEA exhibit greater bias than other ET products in several vegetation types, such as CRO, DNF, ENF, GRA? These vegetation types are the main types on the land surface. Further clarification on the reasons for this discrepancy would enhance the interpretation of the results.*

Re: We added the explanation of the greater bias in CRO, DNF, ENF, GRA at lines 636-666:

"Incoming shortwave radiation from ERA5-Land is employed to derive the available energy for vegetation coverage and bare soil (Eq. 14 and 15), which are the main parameters for calculating daily ET (Eq. 11). While ERA5-Land is widely utilized as a reanalysis dataset, offering near-real-time land variables by integrating model data with global observations based on physical laws. However, the accuracy of shortwave radiation from ERA5-Land seems compromised in savannas (Figure 3) due to the challenges associated with simulating radiation transmission under land-use changes and aerosol pollution from natural or anthropogenic sources.

Air temperature is an important parameter in determining the daily evaporation fraction of bare soil (Appendix B), canopy surface resistance, aerodynamic resistance of the bare soil (Appendix D) and atmospheric emissivity (Appendix E), available energy for vegetation coverage and bare soil (Eq. 14 and 15). Since air temperature is not measured directly by satellites, many other ET product use therefore ground observations, land model or reanalysis data. In contrast, VISEA derives the air temperature from the negative linear relationship between vegetation index (VI) and surface temperature (Ts) using the VI-Ts method (section 2.1.3). It gives very good results under grass land, open shrubland and woody savannas landcover types, as shown in Figure 4. As previously explained, the VI-Ts method relies on the negative linear correlation between the Vegetation Index (VI) and surface temperature (Ts) within a 5 × 5 grid. Therefore, both the variance of VI values across these grid cells and the negative correlation are essential

for calculating the air temperature. However, in regions where the vegetation index and temperature data in adjacent grid cells show small variations, such as dense forests and bare lands and deserts. Also, in regions with freezing temperatures, the VI-T$_S$ method does perform well, because warmer temperature is related to increased vegetation, opposite the other regions, where there is a negative.

Another bias source of the VISEA model is the uncertainties of daily net radiation, notably originating from input downward shortwave radiation from ERA5-Land (Figure 2) and VI-Ts estimated air temperature (Figure 4). The energy budget equation (Eq. 11) and these two figures indicate that net radiation shows more uncertainties than shortwave radiation and air temperature. At the same time, assuming a linear relationship between cloud coverage (Eq. 12 and 13) and the calculation of downwards longwave radiation (Eq. 14 and 15) may be an oversimplification that could introduce uncertainties. Since available energy for evapotranspiration (ET) depends on net radiation (Eq. 16 and 17), addressing these uncertainties is crucial for enhancing overall model accuracy (Brutsaert, 1975; Huang et al., 2023). Future refinements will contribute to a more precise daily net radiation estimation within the VISEA model."

4.      Figure 8: Why VISEA presents an obvious higher ET values compared to other products in most tropical areas of South America and Africa?

Re: we have addressed this at lines 667-676, providing a detailed explanation for the higher ET values calculated by VISEA in tropical areas of South America and Africa.

"The VISEA model calculates ET primarily based on vegetation coverage, utilizing it as an indirect constraint to estimate evapotranspiration. However, this model does not directly incorporate variables related to water availability, which is a critical factor in ET processes. In tropical regions, where there is an abundance of solar radiation (available energy), the model tends to overestimate ET due to its emphasis on vegetation coverage without adequately accounting for the actual water available for evapotranspiration. This methodology, while effective in capturing the influence of vegetation on ET under varied conditions, can lead to overestimations in areas where energy availability significantly exceeds water availability, typical of many tropical regions. Our analysis and subsequent discussion aim to highlight this characteristic of the VISEA model, acknowledging its implications for ET estimations in such energy-rich, water-variable environments."

*5.       Section 2.1 should avoid repetition if the different modules and steps of these methods have been clearly described in previous published papers, ensuring that the current manuscript offers new and valuable information.*

Re: We have transferred repeated equations (Eq. 2,3, 6-8 and 10-12) to the Appendix A (Eq. 2 and 3) and B (Eq. 6-8 and 10-12). However, we retained a selection of essential equations within the main text (Eq. 1, 4, 5, 9, 13-22). This approach is intended to facilitate a comprehensive understanding of the calculation process for readers who may not be familiar with our previous work, while simultaneously streamlining the manuscript to emphasize new and valuable insights.

*6.       Line 64: "these models often have limited spatial resolutions, making them less effective...", I do not think this is the truth. The ERA5 reanalysis, which combines climate model simulation and observational data, also produced latent heat flux (ET in energy units) with a delay of six days.*

Re: we have rewritten this sentence at lines 64-70:
"While these models such as ERA5 reanalysis offer near-real-time latent heat flux (ET in energy units) with a delay of just six days, they typically feature coarser spatial resolutions, often 0.1 degrees or more. This level of resolution may limit their effectiveness for detailed assessments of drought conditions and the optimization of water resource allocation."

*7.       Line 65-68: It would be valuable to highlight the advantages of satellite remote sensing-based ET estimates compared to climate model simulation.*

Re: we have highlighted the advantages of satellite remote sensing-based ET estimates compared to climate model simulation at lines 71-76:
"Satellite remote sensing-based ET estimates outperform climate model simulations by offering high spatial resolution for detailed water use analysis, near-real-time data for prompt environmental response, and global coverage for comprehensive water cycle studies. These estimates rely on direct observations, enhancing accuracy, especially where ground data are sparse, and allow for the dynamic monitoring of

land and vegetation changes. This capability underscores their importance in water resource management and climate research, complementing the broader perspectives provided by climate models."

*8.        Did the authors perform energy closure correction or validation for the FLUXNET observational data?*

Re: In our evaluation using FLUXNET observational data, we benefited from the extensive efforts of FLUXNET in addressing energy closure concerns. Specifically, the high-quality and gap-filling data from the 212 globally distributed flux towers. FLUXNET has implemented measures to perform energy closure corrections and validations to enhance the reliability of the observational data. For detailed information on these procedures, we refer to these publications: titled 'FLUXNET: A New Tool to Study the Temporal and Spatial Variability of Ecosystem-Scale Carbon Dioxide, Water Vapor, and Energy Flux Densities' (Baldocchi et al., 2001). And Pastorello, G., Trotta, C., Canfora, E. et al. The FLUXNET2015 dataset and the ONEFlux processing pipeline for eddy covariance data. Sci Data 7, 225 (2020). https://doi.org/10.1038/s41597-020-0534-3 and Wang, R., Li, L., Gentine, P., Zhang, Y., Chen, J., Chen, X., Chen, L., Ning, L., Yuan, L., and Lü, G.: Recent increase in the observation-derived land evapotranspiration due to global warming, Environ. Res. Lett., 17, 024020, https://doi.org/10.1088/1748-9326/ac4291, 2022.

We added the at lines 316-322

"In our evaluation using FLUXNET observational data, we leveraged FLUXNET's diligent efforts in addressing energy closure concerns. Specifically, FLUXNET has implemented rigorous measures for energy closure corrections and validations, thereby enhancing the reliability of the observational data from the 212 globally distributed flux towers (Pastorello et al., 2020; Baldocchi et al., 2001; Wang et al., 2022), We selected data spanning the period from 2001 to 2015 and excluded sites where ERA5-Land downward shortwave radiation was zero."

Here added the full reference:
Baldocchi, D., Falge, E., Gu, L., Olson, R., Hollinger, D., Running, S., Anthoni, P., Bernhofer, C., Davis, K., Evans, R., Fuentes, J., Goldstein, A., Katul, G., Law, B., Lee, X., Malhi, Y., Meyers, T., Munger, W., Oechel, W., U, K. T. P., Pilegaard, K., Schmid, H. P., Valentini, R., Verma, S., Vesala, T., Wilson, K., and Wofsy, S.: FLUXNET: A New Tool to Study the Temporal and Spatial Variability of Ecosystem-Scale Carbon Dioxide,

Water Vapor, and Energy Flux Densities, Bulletin of the American Meteorological Society, 82, 2415–2434, https://doi.org/10.1175/1520-0477(2001)082<2415:FANTTS>2.3.CO;2, 2001.

Pastorello, G., Trotta, C., Canfora, E., Chu, H., Christianson, D., Cheah, Y.-W., Poindexter, C., Chen, J., Elbashandy, A., Humphrey, M., Isaac, P., Polidori, D., Reichstein, M., Ribeca, A., van Ingen, C., Vuichard, N., Zhang, L., Amiro, B., Ammann, C., Arain, M. A., Ardö, J., Arkebauer, T., Arndt, S. K., Arriga, N., Aubinet, M., Aurela, M., Baldocchi, D., Barr, A., Beamesderfer, E., Marchesini, L. B., Bergeron, O., Beringer, J., Bernhofer, C., Berveiller, D., Billesbach, D., Black, T. A., Blanken, P. D., Bohrer, G., Boike, J., Bolstad, P. V., Bonal, D., Bonnefond, J.-M., Bowling, D. R., Bracho, R., Brodeur, J., Brümmer, C., Buchmann, N., Burban, B., Burns, S. P., Buysse, P., Cale, P., Cavagna, M., Cellier, P., Chen, S., Chini, I., Christensen, T. R., Cleverly, J., Collalti, A., Consalvo, C., Cook, B. D., Cook, D., Coursolle, C., Cremonese, E., Curtis, P. S., D'Andrea, E., da Rocha, H., Dai, X., Davis, K. J., Cinti, B. D., Grandcourt, A. de, Ligne, A. D., De Oliveira, R. C., Delpierre, N., Desai, A. R., Di Bella, C. M., Tommasi, P. di, Dolman, H., Domingo, F., Dong, G., Dore, S., Duce, P., Dufrêne, E., Dunn, A., Dušek, J., Eamus, D., Eichelmann, U., ElKhidir, H. A. M., Eugster, W., Ewenz, C. M., Ewers, B., Famulari, D., Fares, S., Feigenwinter, I., Feitz, A., Fensholt, R., Filippa, G., Fischer, M., Frank, J., Galvagno, M., et al.: The FLUXNET2015 dataset and the ONEFlux processing pipeline for eddy covariance data, Sci Data, 7, 225, https://doi.org/10.1038/s41597-020-0534-3, 2020.

Wang, L., Liu, H., Chen, D., Zhang, P., Leavitt, S., Liu, Y., Fang, C., Sun, C., Cai, Q., Gui, Z., Liang, B., Shi, L., Liu, F., Zheng, Y., and Grießinger, J.: The 1820s Marks a Shift to Hotter-Drier Summers in Western Europe Since 1360, Geophysical Research Letters, 49, e2022GL099692, https://doi.org/10.1029/2022GL099692, 2022.

*9.      Line 197: Is this the truth? It may be necessary to provide additional context or references to support the claim that the daily-scale G is approximately 0 and can be ignored.*

Re: we have added the related references to support the claim that the daily-scale G is approximately 0 and can be ignored at lines 192-193:

"$G \approx 0$ on a daily basis (Fritschen and Gay, 1979; Nishida et al., 2003; Tang et al., 2009)"

Here added the full references:

Fritschen, L. J. and Gay, L. W.: Soil Heat Flux, in: Environmental Instrumentation, edited by: Fritschen, L. J. and Gay, L. W., Springer, New York, NY, 86–92, https://doi.org/10.1007/978-1-4612-6205-3_4, 1979.

Nishida, K., Nemani, R. R., Running, S. W., and Glassy, J. M.: An operational remote sensing algorithm of land surface evaporation, Journal of Geophysical Research: Atmospheres, 108, https://doi.org/10.1029/2002JD002062, 2003.

Tang, Q., Peterson, S., Cuenca, R. H., Hagimoto, Y., and Lettenmaier, D. P.: Satellite-based near-real-time estimation of irrigated crop water consumption, Journal of Geophysical Research: Atmospheres, 114, https://doi.org/10.1029/2008JD010854, 2009.

*10.     Considering the availability of more recent products, such as Jung's FLUXCOM (2019), and the potential impact of sensor developments, the choice of using older data sources, such as AVHRR (2001-2006), should be justified. Comparisons with older data may be influenced by advancements in sensor and computer technologies.*

Re: we opted to use the FLUXCOM data instead of the older GBAF and the last update ET from GLEAM data (covering the period from 2001 to 2022), as depicted in Table 2. However, utilizing AVHRR data allows us to conduct meaningful comparisons between the ET calculated by MODIS and AVHRR data in a similar calculation progress. We believe this comparison will provide valuable insights into any differences influenced by sensor and technology advancements over time.

**Table 2.** The five global girded ET products and one precipitation product used for comparison with our near-real-time global daily terrestrial ET estimates.

| Product name | Spatial/Temporal resolution | Time period | Theory |
|---|---|---|---|
| GLEAM | 0.25°/Monthly | 2001-2022 | Priestly-Taylor Equation |
| FLUXCOM | 0.5°/Monthly | 2001-2016 | Machine learning |
| MOD16 | 0.05°/Monthly | 2001-2014 | Penman-Monteith Equation |
| AVHRR | 1°/Monthly | 2001-2006 | Improved Penman-Monteith Equation |
| PML | 0.05°/8-day | 2003-2018 | Penman-Monteith Equation and a diagnostic biophysical model |
| GPCC | 0.25°/Monthly | 2001-2019 | in-situ observations |
| GPC | 0.5°/Daily | 08/28/2022-09/01/2022 | Global Unified Gauge-Based Analysis of Daily Precipitation |

[Figure]

**Figure 7.** Taylor Diagrams comparing monthly measurements of (a) VISEA, GLEAM (b), FLUXCOM (c), AVHRR (d), MOD16 (e), and PML (f) with 150 flux towers (labeled as Obv) in different IGBP land cover types. The diagrams

display the Normalized Standard Deviation (represented by red circles), Correlation Coefficient (shown as green lines), and Centred Root-Mean-Square (depicted as blue circles).

11.    *Line 409: Please briefly explain the method used to estimate Rn in VISEA. I suspect it is fundamentally different from ERA5_Rd. Please clarify the purpose and significance of comparing these two variables.*

Re: ERA5_Rd is the input data of VISEA which was used to calculate Rn. We have explained the method we used to calculated daily net radiation at lines 188-196:

"We used an improved daily available energy $Q$ (W m$^{-2}$) method (Huang et al., 2023) for the vegetation and the bare soil surface is calculated by the energy balance equation:

$$R_n - G = Q \qquad (10)$$

where $R_n$ is the net radiation (W m$^{-2}$), which could be calculated by the land surface energy balance; $G$ is the soil heat flux (W m$^{-2}$), $G \approx 0$ on a daily basis (Fritschen and Gay, 1979; Nishida et al., 2003; Tang et al., 2009),

$$R_n^d = \left(1 - albedo^d\right)R_d^d - \varepsilon_s^d \sigma T_s^{d\,4} + \left(1 + Cloud^d\right)\varepsilon_a^d \sigma T_a^{d\,4} \qquad (11)$$

Where $albedo^d$ is the daily albedo of the soil surface; $R_d^d$ is daily incoming shortwave radiation (W m$^{-2}$), obtained the ERA5_Land shortwave radiation (we called ERA5_Rd); $\varepsilon_s^d$ and $\varepsilon_a^d$ are the daily emissivity of land surface and atmosphere; different from the former study provided by Huang et al., (2021), which set we $\varepsilon_s^d$ and $\varepsilon_a^d$ equal, we calculated the $\varepsilon_a^d$ by Appendix B flowing study of Brutsaert, (1975) and Wang and Dickinson(2013), $\varepsilon_s^d$ can be retried by MOD11C1; $\sigma$ is the Stefan-Boltzmann constant; $T_a^d$ is the daily near surface air temperature (K); $T_s^d$ is the daily surface temperature (K). "

12.    *Line 548: "actual ET measurement"?*

Re: We have modified as "actual ET estimation" at lines 615-616

13.    *Line 605: Please why the VISEA approach performs better than other products at DNF? I am not sure, but the bias and RMSE values of VISEA at DNF are larger than other products (see Table 3)?*

Re: we have rewritten this paragraph and explained the reason of the VISEA approach has higher bias and RMSE at DNF at lines 677-686:

"Furthermore, the VISEA model exhibits a tendency to underestimate ET in colder regions within the 60°N to 90°N latitude range, such as the western territories of Canada. This underestimation is primarily due to the model's inability to incorporate evaporation from frozen surfaces into its ET calculations. These discrepancies arise from several factors: inaccuracies in the ERA5-Land shortwave radiation data (illustrated in Figure 3), the misapplication of the VI-Ts method (explained in Figure 4), and the uncertainties in daily net radiation (depicted in Figure 5). Designed to amalgamate bare soil and full vegetation coverage as depicted in Equation 1, the VISEA model encounters difficulties in accurately estimating ET at higher latitudes, especially in conditions of reduced solar radiation. These challenges are predominantly linked to the uncertainties associated with ERA5-Land shortwave radiation data, further compounded by increased cloudiness levels in these regions, as highlighted by Babar et al. (2019). Such uncertainties have a substantial impact on the model's performance at higher latitudes, affecting its reliability in these conditions."

---

## Author Response (AR2)

**Response to Liu Mingliang**

essd-2023-495

Title: Satellite-based Near-Real-Time Global Daily Terrestrial Evapotranspiration Estimates

Author(s): Lei Huang et al.

MS type: Data description paper

In the reply, the reviewers' comments are in *italics*, our response is in normal text, and quotes from the manuscript are in **blue**.
* * *
*I still doubt the advantage of this study from other earlier data products. The authors claim that this new products can estimate ET in within a week rather than other products delays of more than two weeks (L617-619), while this new data product is not near-real time at all (only covers 2001-2022) and there is no operational platform provides this (near-real time) service. The descriptions on the equations are still confusing in many places (the organization of equations in the text and appendix need rearrangement). Some equation numbers in Fig 1 are wrong. The comparisons (and evaluations) between ET products from this study and GPCC precipitation is not direct and no robust conclusions could produced from this comparison (although authors claim that there are matched spatial pattern as described in L730-734, other models or products could have the same results). This paper did a good job in comparing this data products with others and observations from fluxnet, while it could do a better job to fix the problems (i.e. decrease the biases) since the method itself (VISEA) had already been developed well before this paper.*

Re: we thank the reviewers!

We have updated our ET data product up to March 20, 2024. All current data can be accessed online at https://doi.org/10.11888/Terre.tpdc.300782. We are committed to continuously updating this dataset, ensuring that the latest ET data will be consistently and promptly made available. Further, we have included a description of the operational platform that supports this dataset at lines 735-740:

The VISEA code for calculating daily ET is written in C and can be executed on Windows 10 using an Intel(R) Core (TM) i7-8565U CPU @ 1.80GHz, 1.99 GHz, 16.0 GB RAM with Visual Studio 2019, or compatible platforms. Additionally, it can run on high-performance computing servers equipped with an Intel(R) Xeon(R) CPU E5-2680 in a CentOS environment. The system is scalable, supporting configurations ranging from 20 nodes and 656 CPUs down to fewer nodes and CPUs as required.

We have reorganized and clarified the descriptions of equations within the text and appendix. These changes can be found at lines 141-178:

**2.1.1 Daily evaporation fraction calculation**

[revised manuscript text omitted]

We have corrected the equation numbers in Figure 1 to ensure accuracy and clarity.

[Figure]

**Figure 1.** Schematic of VISEA algorithm. The ovals in the top row are the databases, and the square boxes are the algorithms, and parallelograms are the parameters. The numbers in the parenthesis are the equation to determine the parameters.

In our previous studies (Huang et al., 2021 & 2023b), we have continually focused on enhancing the accuracy of the VISEA model, specifically by improving the calculations of daily evaporation fraction and daily net radiation in China. Our current efforts aim to integrate these improvements, expand our study to a global territorial scale, and evaluate VISEA's performance internationally. We are also preparing to publish both the refined algorithms and the global-scale ET data.

Our primary goal is to demonstrate the model reliability on a global scale, and we are committed to ongoing enhancements. Future updates will specifically target reducing biases and further improving the overall performance

*L44-46: this claim is problematic since one (i.e. GPCC) is precipitation while another one is ET, so they are difference things even though they have high correlations.*

Re: We understand the concern about comparing fundamentally different variables. However, the purpose of this comparison is to validate the spatial distribution of our estimated ET by showing its reasonable correlation with known precipitation patterns. Similar methodologies have been employed in several notable studies, for example: Mu et al., 2007 evaluated the MOD16 data product by comparing precipitation distributions with estimated ET in their Figures 8 and 9 as shown below:

[Figure]

Fig. 8. (a) Annual total precipitation and (b) annual total MODIS GPP versus annual total ET (driven by GMAO meteorological data) in 2001. The solid line in (a) represents that the ratio of ET to precipitation is 1.0.

[Figure]

Fig. 9. (a) Global precipitation (precip) in 2001 with a maximum of 7588 mm/yr and an average of 780±686 mm/yr over land. (b) The difference between annual precipitation and annual ET (driven by GMAO meteorological data) in 2001, with a maximum of 4476 mm/yr and an average of 586±568 mm/yr. Vegetated regions are shown in color, and the regions in white are barren or sparsely vegetated areas and non-vegetated areas, including water bodies, snow and ice, and urban areas.

Another example is by Zhang et al., 2019 who compared the distribution of estimated ET and Gross Primary Production (GPP) in their Figure 9.

[Figure]

Fig. 9. PML-V2 global average maps of (a) mean annual ET and (b) mean annual GPP for 2003–2017, and their distributions averaged across 1/12° resolution latitudinal bands. These maps are generated at the 1/12° spatial resolution by averaging all 500 m pixels. Water bodies and permanent ice surfaces are excluded for latitudinal averages.

These comparisons demonstrate that the spatial patterns of estimated ET align with other environmental variables and support the validity of the ET estimations.

References:

Mu, Q., Heinsch, F., Zhao, M., and Running, S.: Development of a global evapotranspiration algorithm based on MODIS and global meteorology data, Remote Sensing of Environment, 111, 519–536, https://doi.org/10.1016/J.RSE.2007.04.015, 2007.

Zhang, Y., Kong, D., Gan, R., Chiew, F. H. S., McVicar, T. R., Zhang, Q., and Yang, Y.: Coupled estimation of 500 m and 8-day resolution global evapotranspiration and gross primary production in 2002–2017, Remote Sensing of Environment, 222, 165–182, https://doi.org/10.1016/j.rse.2018.12.031, 2019.

*L168: add "temperature" after "air".*

Re: We have added temperature after "air" at lines 139-140

In the next section, we will detail how VISEA calculates the daily $EF$, and $Q$ in Eq. 3, daily air temperature and daily land surface temperature.

*The connections between equ. 6 & 7 is not clear, equation B2 might be inserted between these two equations not need explanations.*

Re: We have repositioned Equations B1 to B3 between Equations 6 and 7, as they are closely interconnected and provide necessary context for understanding the relationship between these equations at lines 142-156:

Combining Eq. 1, 2 and 3, we calculated the instantaneous evaporation fraction, $EF^i$ as:

$$EF^i = f_{veg} \frac{Q^i_{veg}}{Q^i} EF^i_{veg} + (1 - f_{veg}) \frac{Q^i_{soil}}{Q^i} EF^i_{soil} \tag{4}$$

$EF^i_{veg}$ and $EF^i_{soil}$ are the instantaneous full vegetation coverage and bare soil $EF$, respectively. $EF^i_{veg}$ can be expressed as a function of instantaneous parameters (Nishida et al., 2003):

$$EF^i_{veg} = \frac{\alpha \Delta^i}{\Delta^i + \gamma(1 + r^i_{c\,veg}/2r^i_{a\,veg})} \tag{5}$$

where $\alpha$ is the Priestley-Taylor parameter, which was set to 1.26 for wet surfaces (De Bruin, 1983); $\Delta^i$ is the instantaneous slope of the saturated vapor pressure, which is a function of the temperature (Pa K$^{-1}$); $\gamma$ is the psychometric constant (Pa K$^{-1}$); $r^i_{c\,veg}$ is the instantaneous surface resistance of the vegetation canopy (s m$^{-1}$); $r^i_{a\,veg}$ is the instantaneous aerodynamics resistance of the vegetation canopy (s m$^{-1}$). $EF^i_{soil}$ was expressed by Nishida et al. (2003) as a function of the instantaneous soil temperature and the available energy based on the energy budget of the bare soil:

$$EF^i_{soil} = \frac{T^i_{soil\,max} - T^i_{soil}}{T^i_{soil\,max} - T^i_a} \frac{Q^i_{soil0}}{Q^i_{soil}} \tag{6}$$

where $T^i_{soil\,max}$ is the instantaneous maximum possible temperature at the surface reached when the land surface is dry (K), $T^i_{soil}$ is the instantaneous temperature of the bare soil (K), $T^i_a$ is the instantaneous air temperature, $Q^i_{soil0}$ is the instantaneous available energy for bare soil when $T^i_{soil}$ is equal to $T^i_a$ (W m$^{-2}$).

*Fig. 8: replace "CPCC" with "GPCC" and suggest add "precipitation" after "GPCC" since only this one is not ET but precipitation.*

Re: we have replaced "CPCC" with "GPCC" and added "precipitation" after "GPCC" in Figure 8.

[Figure]

**Figure 8.** The spatial distribution of the multi-year average (a-g), the zonal mean (h) and inter-annual variation (i) of (a) GPCC precipitation (2001-2019), (b) VISEA (2001-2020), (c) GLEAM (2001-2020), (d) FLUXCOM (2001-2016), (e) AVHRR (2001-2006), (f) MOD16 (2001-2014) and (g) PML (2003-2018) ET data.

*L178: incomplete sentence.*

Re: We have rewritten this sentence at line 166*:*

According to Pereira (2004), the calculation details of $\Omega$ and $\Omega^*$ are presented in Appendix B.

*Fig.1: the equation numbers are not consistent with the text.*

Re: We have corrected the equation numbers of Figure 1.

[Figure]

**Figure 1.** Schematic of VISEA algorithm. The ovals in the top row are the databases, and the square boxes are the algorithms, and parallelograms are the parameters. The numbers in the parenthesis are the equation to determine the parameters.

*L185: The "Thus" connection is not clear. Equ. 9: how instantaneous Q, Qsoil and Qveg (and its defination in equ. 8) are calculated?*

Re: The sentence has been rewritten as follows at lines 176-178:

The same energy balance equations are used for calculating both instantaneous values $Q^i$, $Q^i_{veg}$ and $Q^i_{soil}$ and daily values $Q^d$, $Q^d_{veg}$ and $Q^d_{soil}$ but with parameters adjusted for each timeframe The details of the calculation for the daily values are outlined below.

*All equations should be consistent and the name of physical variables should be defined, such as the cloud in equ. 12 and Qsoil 0 .*

Re: The definition of $Cloud^d$ in Eq. 15 is added at line 198-200

where $Cloud^d$ is the daily clearness index and $K_t$ is (Chang and Zhang, 2019; Goforth et al., 2002)

$$K_t = \frac{R^d_d}{R^d_a} \tag{14}$$

where $R^d_a$ is the daily extraterrestrial radiation calculated by the FAO (1998).

$Q^i_{soil0}$ is defined at line 156

$Q^i_{soil0}$ is the instantaneous available energy for bare soil when $T^i_{soil}$ is equal to $T^i_a$ (W m$^{-2}$)

*L208-210: I don't know what logic is there*

Re: These sentences have been rewritten at lines 203-211:

According to the land surface energy budget, the daily available energy of vegetation coverage area, $Q^d_{veg}$ and bare soil $Q^d_{soil}$ can be calculated following the study of Huang et al. (2023b):

$$Q^d_{veg} = (1 - albedo^d)R^d_d + (1 + Cloud^d)\varepsilon^d_a \sigma\, T^{d\,4}_a - \varepsilon^d_s \sigma T^{d\,4}_s \tag{15}$$

$$Q^d_{soil} = (1 - C_G)(1 - albedo^d)R^d_d + (1 + Cloud^d)\varepsilon^d_a \sigma\, T^{d\,4}_a - \varepsilon^d_s \sigma T^{d\,4}_s \tag{16}$$

The daily mean air temperature, $T^d_a$ can be extended by a sin and cos function based on the instantaneous air temperature $T^i_a$ which was calculated using the linear correlation between vegetation index (VI) and surface temperature (Ts) method. Thus, $(1 + Cloud^d)\varepsilon^d_a \sigma\, T^{d\,4}_a$ is the daily downward longwave radiation (W m$^{-2}$), and $\varepsilon^d_s \sigma T^{d\,4}_s$ is the daily upward longwave radiation (W m$^{-2}$), where $C_G$ is an empirical coefficient ranging from 0.3 for a wet soil to 0.5 for a dry soil (Idso et al., 1975).

*L401-403: the description is vague since Fig. 3 shows comparisons between fluxnet and ERA, nothing related to MODIS products (besides the land cover).*

Re: Additional information is provided at lines 390-394:

We chose to utilize 0.05° MODIS data for its detailed land surface information, daily time step, and global coverage, which is essential for accurate and near-real-time ET calculations. Although ERA5 data is at a coarser 0.1° resolution, it provides necessary atmospheric inputs that can be effectively interpolated to match the MODIS resolution without significant loss of accuracy. As illustrated in Figures 3 and 4, our tests confirm that this method achieves accurate ET despite the resolution differences.

*L554-590: most of the results are not new and should avoid to say which model "overestimate" or "underestimate" without compare with the ground truth.*

Re: we have rewritten these paragraphs at lines 564-600:

The VISEA ET product demonstrates consistent spatial distribution patterns among the six ET products across various years in terms of annual means (a-g) and latitude zonal means (h). These patterns closely align with the precipitation distribution data from GPCC. Furthermore, VISEA ET also exhibit similar spatial distributions compared to other ET products, particularly in the extremes of the distribution, below the 5$^{th}$ percentile and above the 95$^{th}$ percentile (Figure S6, S7). The highest ET values, approximately 1,500 mm year$^{-1}$, are predominantly in equatorial low-latitude regions with the corresponding high precipitation levels of approximately 2,500 mm year$^{-1}$. These regions include South America (Amazon Basin), Central Africa (Congo Basin), and Southeast Asia (encompassing Indonesia, Malaysia, parts of Thailand, and the Philippines), which have tropical rainforest climates. Remote sensing data support the ET estimates and align with findings from previous studies, such as Chen et al. (2021) and Zhang et al. (2019), who reported that the multi-year average annual ET is nearly 1,500 and the precipitation is approximately 2,500 mm year$^{-1}$. Also, Panagos et al. (2017) report similar multi-year average annual ET and precipitation rates.

In this analysis, barren lands (BAR) such as the Sahara, Arabian, Gobi, and Kalahari deserts, along with large areas of Australia, and snow and ice (SI) regions including significant parts of Canada, Russia, and the Qinghai-Tibet Plateau in China, are characterized by notably low evapotranspiration (ET). These regions typically experience less than 400 mm year$^{-1}$ of annual ET, paralleled by minimal yearly precipitation ranging from 200 to 400 mm year$^{-1}$, according to GPCC data. Comparative ET rates for other land cover types generally range from 400 to 1,400 mm year$^{-1}$, closely following the GPCC precipitation amounts of 600 to 1,600 mm year$^{-1}$.

In regions experiencing moisture-limited evapotranspiration (ET), the scarcity of available water is the primary constraint. Conversely, in areas where sufficient water is available, ET is energy-limited, and factors such as cloud cover or shading restrict the absorption of solar radiation, affecting the evapotranspiration rate. Panel (i) in Figure 8 illustrates inter-annual monthly variations over the past two decades. It shows how VISEA and other satellite-based ET products, alongside GPCC precipitation data, capture the rhythmic patterns of ET. These data reveal distinctive seasonal fluctuations and highlight the significant inter-annual climate variability. Among these products, FLUXCOM consistently shows ET values 10-20 mm month$^{-1}$ higher than those of other ET products. GLEAM and MOD16 exhibit similar ET estimations, closely paralleling each other, as do PML and VISEA. Notably, after 2007, both GLEAM and MOD16 reported higher ET estimations than PML and VISEA in November, December, January, and February. For the same months, PML consistently records lower ET estimations than VISEA.

Analysis across the datasets reveals how ET estimates respond to extreme climate events, providing insights into the variability and resilience of these models. For instance, during the 2011-2012 drought in the Horn of Africa—one of the most severe droughts in recent decades—both ET estimations and GPCC precipitation data showed significant declines. Similarly, the prolonged California drought from 2012 to 2016 also saw a considerable decrease in ET values, aligning with the reduced precipitation levels captured by GPCC.

*Fig.8: need dig into more details on the differences in long-term trends and the variations in the years with extreme climate events.*

Re: we have added more details on the differences in long-term trends and the variations in the years with extreme climate events at lines 590-600:

Among these products, FLUXCOM consistently shows ET values 10-20 mm month$^{-1}$ higher than those of other ET products. GLEAM and MOD16 exhibit similar ET estimations, closely paralleling each other, as do PML and VISEA. Notably, after 2007, both GLEAM and MOD16 reported higher ET estimations than PML and VISEA in November, December, January, and February. For the same months, PML consistently records lower ET estimations than VISEA.

Analysis across the datasets reveals how ET estimates respond to extreme climate events, providing insights into the variability and resilience of these models. For instance, during the 2011-2012 drought in the Horn of Africa—one of the most severe droughts in recent decades—both ET estimations and GPCC precipitation data showed significant declines. Similarly, the prolonged California drought from 2012 to

2016 also saw a considerable decrease in ET values, aligning with the reduced precipitation levels captured by GPCC.

*L591-612: the arguments are statistical weak and the mismatch of spatial (and temporal) pattern between VISEA and precipitation can not support the conclusion that VISEA is better than another model. Authors need discuss the effects of soil moisture on ET, besides the precipitation events.*

Re: This part is modified by adding a discussion on the effects of soil moisture on ET, besides the precipitation events at lines 609-629:

[revised manuscript text omitted]

*L843-845: it's still very strange to claim that VPD and leaf water potential "can be omitted" from the calculation of canopy resistant. Which factor could affect stomata conductance then?*

Re: we have rewritten this sentence at lines 828-836:

In this study, we follow the methodologies originally developed by Tang et al. (2009) and Nishida (2003), with the goal of enhancing the VISEA model to accurately estimate daily scale evaporation fraction and net radiation. These efforts build on earlier work by Huang et al. (2017, 2021, and 2023b) that introduced vapor pressure deficit (VPD) and leaf water potential in calculating canopy resistance. However, comparative analyses between VISEA and other models, such as PML and MOD16—particularly PML, which integrates VPD as a limiting factor in estimating GPP and ET—show that VISEA maintains accuracy without significant biases. It is important to note that none of the ET models in our comparison directly incorporate leaf water potential into their canopy resistance calculations. We are committed to addressing these gaps in our future studies.